# Beyond the Doors of Perception: Vision Transformers Represent Relations Between Objects

**Michael A. Lepori**[1]*    **Alexa R. Tartaglini**[2,3]*
**Wai Keen Vong**[2]    **Thomas Serre**[1]    **Brenden M. Lake**[2]    **Ellie Pavlick**[1]
[1]Brown University    [2]New York University    [3]Stanford University
{michael_lepori,thomas_serre,ellie_pavlick}@brown.edu
alexart@stanford.edu    {waikeen.vong,brenden}@nyu.edu

## Abstract

Though vision transformers (ViTs) have achieved state-of-the-art performance in a variety of settings, they exhibit surprising failures when performing tasks involving visual relations. This begs the question: how do ViTs attempt to perform tasks that require computing visual relations between objects? Prior efforts to interpret ViTs tend to focus on characterizing relevant low-level visual *features*. In contrast, we adopt methods from mechanistic interpretability to study the higher-level visual *algorithms* that ViTs use to perform abstract visual reasoning. We present a case study of a fundamental, yet surprisingly difficult, relational reasoning task: judging whether two visual entities are the same or different. We find that pretrained ViTs fine-tuned on this task often exhibit two qualitatively different stages of processing despite having no obvious inductive biases to do so: 1) a *perceptual* stage wherein local object features are extracted and stored in a disentangled representation, and 2) a *relational* stage wherein object representations are compared. In the second stage, we find evidence that ViTs can sometimes learn to represent abstract visual relations, a capability that has long been considered out of reach for artificial neural networks. Finally, we demonstrate that failures at either stage can prevent a model from learning a generalizable solution to our fairly simple tasks. By understanding ViTs in terms of discrete processing stages, one can more precisely diagnose and rectify shortcomings of existing and future models.

## 1 Introduction

Despite the well-established successes of transformer models (Vaswani et al., 2017) for a variety of vision applications (ViTs; Dosovitskiy et al. (2020)) – notably image generation and classification – there has been comparatively little breakthrough progress on complex tasks involving *relations* between visual entities, such as visual question answering (Schwenk et al., 2022) and image-text matching (Thrush et al., 2022; Liu et al., 2023; Yuksekgonul et al., 2022). One fundamental difference between these tasks is that the former is largely *semantic* – relying on pixel-level image features that correlate with learned class labels – whereas the latter often involves *syntactic* operations – those which are independent of pixel-level features (Ricci et al., 2021; Hochmann et al., 2021). Though the ability to compute over abstract visual relations is thought to be fundamental to human visual intelligence (Ullman, 1987; Hespos et al., 2021), the ability of neural networks to perform such syntactic operations has been the subject of intense debate (Fodor & Pylyshyn, 1988; Marcus, 2003; Chalmers, 1992; Quilty-Dunn et al., 2023; Lake et al., 2017; Davidson et al., 2024).

Much prior work has attempted to empirically resolve whether or not vision networks can implement an abstract, relational operation, typically by behaviorally assessing the model's ability to generalize

---

*Equal contribution.

38th Conference on Neural Information Processing Systems (NeurIPS 2024).

to held-out stimuli (Fleuret et al., 2011; Zerroug et al., 2022; Kim et al., 2018; Puebla & Bowers, 2022; Tartaglini et al., 2023). However, strikingly different algorithms might produce the same model behavior, rendering it difficult to characterize whether models do or do not possess an abstract operation (Lepori et al., 2023a). This problem is exacerbated when analyzing pretrained models, whose opaque training data renders it difficult to distinguish true generalization from memorization (McCoy et al., 2023). In this work, we employ newly-developed techniques from mechanistic interpretability to characterize the *algorithms* learned by ViTs. Analyzing the internal mechanisms of models enables us to more precisely understand how they attempt to implement relational operations, allowing us to more clearly diagnose problems in current and future models when applied to complex visual tasks.

One of the most fundamental of these abstract operations is identifying whether two objects are the same or different. This operation undergirds human visual and analogical reasoning (Forbus & Lovett, 2021; Cook & Wasserman, 2007), and is crucial for answering a wide variety of common visual questions, such as "How many plates are on the table?" (as each plate must be identified as an instance of the same object) or "Are Mary and Bob reading the same book?" (Ricci et al., 2021). Indeed, same-different judgments can be found across the animal kingdom, being successfully captured by bees (Giurfa et al., 2001), ducklings (Martinho III & Kacelnik, 2016), primates (Thompson & Oden, 2000), and crows (Cook & Wasserman, 2007).

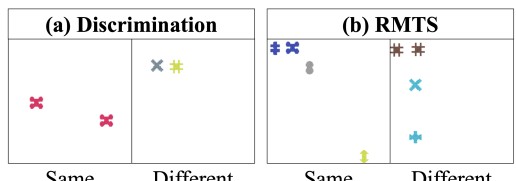

Figure 1: **Two same-different tasks**. **(a) Discrimination**: "same" images contain two objects with the same color and shape. Objects in "different" images differ in at least one of those properties—in this case, both color and shape. **(b) RMTS**: "same" images contain a pair of objects that exhibit the same relation as a display pair of objects in the top left corner. In the image on the left, both pairs demonstrate a "different" relation, so the classification is "same" (relation). "Different" images contain pairs exhibiting different relations.

We analyze ViTs trained on two same-different tasks: an identity discrimination task, which constitutes the most basic instances of the abstract concept of same-different (and is most commonly studied in artificial systems) (Newport, 2021), and a relational match-to-sample task, which requires explicitly representing and manipulating an abstract concept of "same" or "different" (Cook & Wasserman, 2007; Geiger et al., 2023). See Figure 1 for examples of each. We relate the algorithms that models adopt to their downstream behavior, including compositional and OOD generalization[2].

Our main contributions are the following:

1. Inspired by the infant and animal abstract concept learning literature, we introduce a synthetic visual relation match-to-sample (RMTS) task, which assesses a model's ability to represent and compute over the abstract concept of "same" or "different".

2. We identify a processing pipeline within the layers of several – but not all – pretrained ViTs, consisting of a "perceptual" stage followed by a more abstract "relational" stage. We characterize each stage individually, demonstrating that the perceptual stage produces disentangled object representations (Higgins et al., 2018), while the relational stage implements fairly abstract (i.e. invariant to perceptual properties of input images) relational computations – an ability that has been intensely debated since the advent of neural networks (Fodor & Pylyshyn, 1988).

3. We demonstrate that deficiencies in either the perceptual or relational stage can completely prevent models from learning abstract relational operations. By rectifying either stage, models can solve simple relational operations. However, both stages must be intact in order to learn more complex operations.

## 2 Methods

**Discrimination Task**. The discrimination task tests the most basic instance of the same-different relation. This task is well studied in machine learning (Kim et al., 2018; Puebla & Bowers, 2022;

---

[2]Code is available *here*.

Tartaglini et al., 2023) and simply requires a single comparison between two objects. Stimuli in our discrimination task consist of images containing two simple objects (see Figure 1a). Each object may take on one of 16 different shapes and one of 16 different colors (see Appendix A for dataset details). Models are trained to classify whether these two objects are the "same" along both color and shape dimensions or "different" along at least one dimension. Crucially, our stimuli are *patch-aligned*. ViTs tokenize images into patches of $N \times N$ pixels ($N = \{14, 16, 32\}$). We generate datasets such that individual objects reside completely within the bounds of a single patch (for $N = 32$ models) or within exactly 4 patches (for $N = 16$ and $N = 14$ models). Patch alignment allows us to adopt techniques from NLP mechanistic interpretability, which typically assume meaningful discrete inputs (e.g. words). To increase the difficulty of the task, objects are randomly placed within patch boundaries, and Gaussian noise is sampled and added to the tokens (see Appendix A). Models are trained on $6,400$ images.

We evaluate these models on out-of-distribution synthetic stimuli to ensure that the learned relations generalize. Finally, we evaluate a rather extreme form of generalization by generating a "realistic" dataset of discrimination examples using Blender, which include various visual attributes such as lighting conditions, backgrounds, and depth of field (see Appendix A.3).

**Relational Match-to-Sample (RMTS) Task**.    Due to the simplicity of the discrimination task, we also analyze a more abstract (and thus more difficult) iteration of the same-different relation using a relational match-to-sample (RMTS) design. In this task, the model must generate explicit representations of "sameness" and "difference", and then operate over these representations (Martinho III & Kacelnik, 2016; Hochmann et al., 2017). Although many species can solve the discrimination task, animals (Penn et al., 2008) and children younger than 6 (Hochmann et al., 2017; Holyoak & Lu, 2021) struggle to solve the RMTS task. Stimuli in the RMTS task contain 4 objects grouped in two pairs (Figure 1b). The "display" pair always occupies patches in the top left of the image. The "sample" pair can occupy any other position. The task is defined as follows: for each pair, produce a same-different judgment (as in the discrimination task). Then, compare these judgments—if both pairs exhibit the same intermediate judgment (i.e., both pairs exhibit "same" *or* both pairs exhibit "different"), then the label is "same". Otherwise, the label is "different." Objects are identical to those in the discrimination task, and they are similarly patch-aligned.

**Models**.    Tartaglini et al. (2023) demonstrated that CLIP-pretrained ViTs (Radford et al., 2021) can achieve high performance in generalizing the same-different relation to out-of-distribution stimuli when fine-tuned on a same-different discrimination task. Thus, we primarily focus our analysis on this model[3]. In later sections, we compare CLIP to additional ViT models pretrained using DINO (Caron et al., 2021), DINOv2 (Oquab et al.), masked auto encoding (MAE; He et al. (2022)) and ImageNet classification (Russakovsky et al., 2015; Dosovitskiy et al., 2020) objectives. We also train a randomly-initialized ViT model on each task (From Scratch). All models are fine-tuned on either a discrimination or RMTS task for 200 epochs using the `AdamW` optimizer with a learning rate of 1e-6. We perform a sweep over learning rate schedulers (`Exponential` with a decay rate of $0.95$ and `ReduceLROnPlateau` with a patience of $40$). Models are selected by validation accuracy, and test accuracy is reported. Our results affirm and extend those presented in Tartaglini et al. (2023)—CLIP and DINOv2-pretrained ViTs perform extremely well on both tasks, achieving $\geq 97\%$ accuracy on a held-out test set. Appendix B presents results for all models.

## 3    Two-Stage Processing in ViTs

We now begin to characterize the internal mechanisms underlying the success of the CLIP and DINOv2 ViT models. In the following section, we cover how processing occurs in two distinct stages within the model: a *perceptual* stage, where the object tokens strongly attend to other tokens within the same object, and a *relational* stage, where tokens in one object attend to tokens in another object (or pair of objects).

**Methods — Attention Pattern Analysis**.    We explore the operations performed by the model's attention heads, which "read" from particular patches and "write" that information to other patches (Elhage et al., 2021). In particular, we are interested in the flow of information *within* individual objects, *between* the two objects, and (in the case of RMTS) *between two pairs* of objects. We refer to attention heads that consistently exhibit within-object patterns across images as *local* attention

---

[3]Both B/16 and B/32 versions. Results for the B/32 variant are presented in Appendix C.

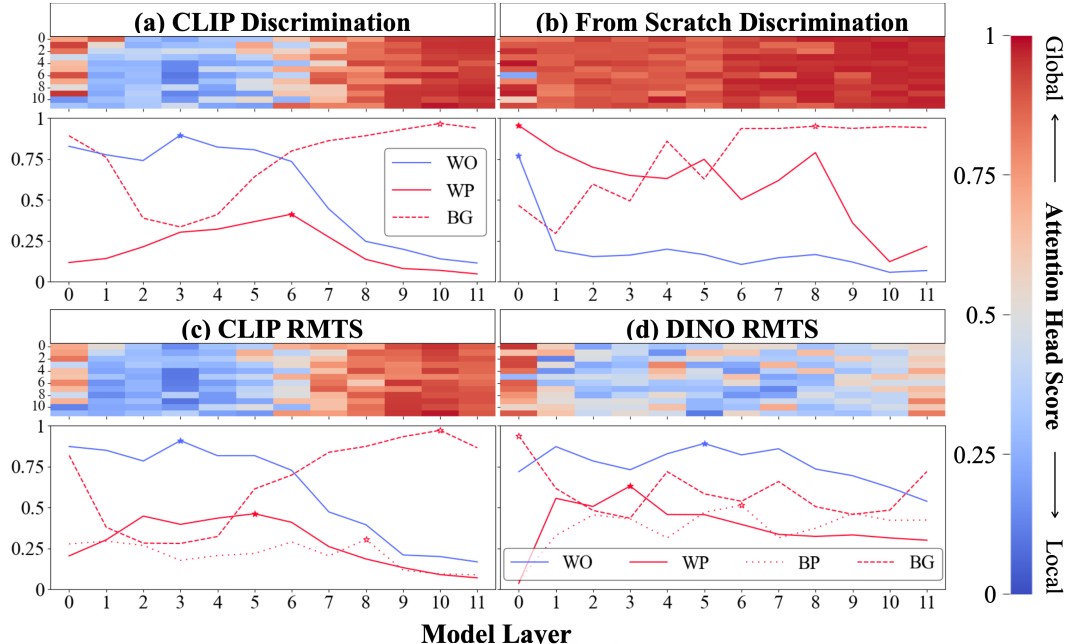

Figure 2: **Attention Pattern Analysis**. **(a) CLIP Discrimination**: The heatmap *(top)* shows the distribution of "local" (blue) vs. "global" (red) attention heads throughout a CLIP ViT-B/16 model fine-tuned on discrimination (Figure 1a). The $x$-axis is the model layer, while the $y$-axis is the head index. Local heads tend to cluster in early layers and transition to global heads around layer 6. For each layer, the line graph *(bottom)* plots the maximum proportion of attention across all 12 heads from object patches to image patches that are 1) within the same object (within-object=**WO**), 2) within the other object (within-pair=**WP**), or 3) in the background (**BG**). The stars mark the peak of each. WO attention peaks in early layers, followed by WP, and finally BG. **(b) From Scratch Discrimination**: We repeat the analysis in (a). The model contains nearly zero local heads. **(c) CLIP RMTS**: We repeat the analysis for a CLIP model fine-tuned on RMTS (Figure 1b). *Top*: Our results largely hold from (a). *Bottom*: We track a fourth attention pattern—attention *between* pairs of objects (between-pair=**BP**). We find that WO peaks first, then WP, then BP, and finally BG. This accords with the hierarchical computations implied by the RMTS task. **(d) DINO RMTS**: We repeat the analysis in (c) for a DINO model and find no such hierarchical pattern.

heads and heads that attend to other tokens *global* attention heads. To classify an attention head as local or global, we score the head from 0 to 1, where values closer to 0 indicate local operations and values closer to 1 indicate global operations. To compute the score for an individual head, we collect its attention patterns on 500 randomly selected "same" and "different" images (1,000 images total). Then, for each object in a given image, we compute the proportion of attention from the object's patches to any other patches that do not belong to the same object (excluding the CLS token)—this includes patches containing the other object(s) and non-object background tokens.[4] This procedure yields two proportions, one for each object in the image. The attention head's score for the image is the maximum of these two proportions. Finally, these scores are averaged across the images to produce the final score.

**Results**. Attention head scores for CLIP ViT-B/16 fine-tuned on the discrimination and RMTS tasks are displayed in the heatmaps of Figure 2a and 2c, respectively. The first half of these models is dominated by attention heads that most often perform local operations (blue cells). See Appendix F for examples of attention patterns. In the intermediate layers, attention heads begin to perform global operations reliably, and the deeper layers of the model are dominated by global heads. The prevalence of these two types of operations clearly demarcates two processing stages in CLIP ViTs: a *perceptual* stage where within-object processing occurs, followed by a *relational* stage where

---

[4]We include attention from object to non-object tokens because we observe that models often move object information to a set of background register tokens (Darcet et al., 2023). See Appendix F.

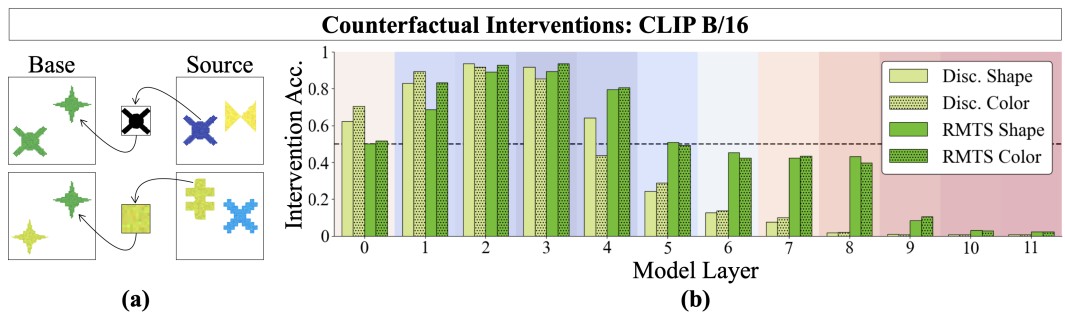

Figure 3: **(a) Interchange interventions**: The base image exhibits the "different" relation, as the two objects differ in either shape *(top)* or color *(bottom)*. An interchange intervention extracts {shape, color} information from the intermediate representations generated by the same model run on a different image (source), then patches this information from the source image into the model's intermediate representations of the base image. If successful, the intervened model will now return "same" when run on the base image. DAS is optimized to succeed at interchange interventions. **(b) Disentanglement Results**: We report the success of interchange interventions on shape and color across layers for CLIP ViT-B/16 fine-tuned on either the discrimination or RMTS task. We find that these properties are disentangled early in the model—one property can be manipulated without interfering with the other. The background is colored according to the heatmap in Figure 2a, where blue denotes local heads and red denotes global heads.

between-object processing occurs. These stages are explored in further depth in Sections 4 and 5 respectively.[5] We also find similar two-stage processing when evaluating on a discrimination dataset that employs realistic stimuli, suggesting that the patterns observed on our synthetic stimuli are robust and transferable (see Appendix A.3).

The line charts in Figure 2 show maximal scores of each attention head type (local and global) in each layer. Since values closer to $0$ indicate local (i.e., *within-object*; WO in Figure 2) heads by construction, we plot these values subtracted from $1$. The global attention heads are further broken down into two subcategories for the discrimination task: *within-pair* attention heads, whereby the tokens of one object attend to tokens associated with the object it is being compared to (WP in Figure 2), and *background* attention heads, whereby object tokens attend to background tokens (BG in Figure 2). We add a third subcategory for RMTS: *between-pair* attention heads, which attend to tokens in the other pair of objects (e.g., a display object attending to a sample object; BP in Figure 2). For both tasks, objects strongly attend to themselves throughout the first six layers, with a peak at layer 3. Throughout this perceptual stage, *within-pair* heads steadily increase in prominence and peak in layer 6 (discrimination) or 5 (RMTS). In RMTS models, this *within-pair* peak is followed by a *between-pair* peak, recapitulating the expected sequence of steps that one might use to solve RMTS. Notably, the *within-pair* (and *between-pair*) peaks occur precisely where an abrupt transition from perceptual operations to relational operations occurs. Around layer 4, object attention to a set of background tokens begins to increase; after layer 6, object-to-background attention accounts for nearly all outgoing attention from object tokens. This suggests that processing may have moved into a set of register tokens (Darcet et al., 2023).

Notably, this two-stage processing pipeline is not trivial to learn—several models, including a randomly initialized model trained on the discrimination task and a DINO model trained on RMTS (Figure 2b and d) fail to exhibit any obvious transition from local to global operations (See Appendix E for results from other models). However, we do find this pipeline in DINOv2 and ImageNet pretrained models (See Appendix D and E). We note that this two-stage processing pipeline loosely recapitulates the processing sequence found in biological vision systems: image representations are first formed during a feedforward sweep of the visual cortex, then feedback connections enable relational reasoning over these representations Kreiman & Serre (2020).

---

[5]We note that this attention pattern analysis is conceptually similar to Raghu et al. (2021), which demonstrated a general shift from local to global attention in ViTs. However, our analysis defines local vs. global heads in terms of objects rather than distances between patches.

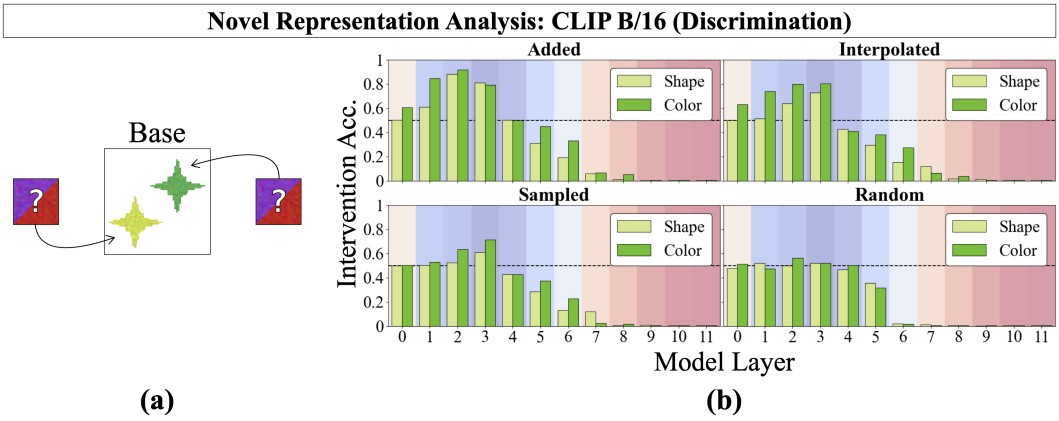

Figure 4: **(a) Novel Representations Analysis**: Using trained DAS interventions, we can inject any vector into a model's shape or color subspaces, allowing us to test whether the same-different operation can be computed over arbitrary vectors. We intervene on a "different" image—differing only in its color property—by patching a novel color (an interpolation of red and black) into *both* objects in order to flip the decision to "same". **(b) Discrimination Results**: We perform novel representations analysis using four methods for generating novel representations: 1) *adding* observed representations, 2) *interpolating* observed representations, 3) per-dimension *sampling* using a distribution derived from observed representations, and 4) sampling *randomly* from a normal distribution $\mathcal{N}(0, 1)$. The model's same-different operation generalizes well to vectors generated by adding (and generalizes somewhat to interpolated vectors) in early layers but not to sampled or random vectors. The background is colored according to the heatmap in Figure 2a (blue=local heads; red=global heads).

## 4 The Perceptual Stage

Attention between tokens is largely restricted to other tokens within the same object in the perceptual stage, but to what end? In the following section, we demonstrate that these layers produce disentangled local object representations which encode shape and color. These properties are represented in separate linear subspaces within the intermediate representations of CLIP and DINOv2-pretrained ViTs.

**Methods — DAS**. Distributed Alignment Search (DAS) (Geiger et al., 2024; Wu et al., 2024c) is used to identify whether particular variables are causally implicated in a model's computation[6]. Given a neural network $M$, hypothesized high-level causal model $C$, and high-level variable $v$, DAS attempts to isolate a linear subspace $s$ of the residual stream states generated by $M$ that represents $v$ (i.e. $s$ takes on a value $s_1$ to represent $v_1$, $s_2$ to represent $v_2$, and so on). The success of DAS is measured by the success of *counterfactual interventions*. If $C(v_1) = y_1$ and $C(v_2) = y_2$, and $M(x) = y_1$ for some input $x$, does replacing $s_1$ with $s_2$ change the model's decision to $y_2$?

Concretely, $M$ corresponds to our pretrained ViT, and a high-level causal model for the discrimination task can be summarized as follows: 1) Extract `shape`$_1$ and `color`$_1$ from `object`$_1$, repeat for `object`$_2$; 2) Compare `shape`$_1$ and `shape`$_2$, compare `color`$_1$ and `color`$_2$; 3) Return `same` if both comparisons return `same`, otherwise return `different`. Similarly, we can define a slightly more complex causal model for the RMTS task. We use this method to understand better the object representations generated by the perceptual stage. In particular, we try to identify whether shape and color are disentangled (Higgins et al., 2018) such that we could edit `shape`$_1 \longrightarrow$ `shape`$_1'$ without interfering with either `color` property (See Figure 3a). For this work, we use a version of DAS where the subspace $s$ is found by optimizing a differentiable binary mask and a rotation matrix over model representations (Wu et al., 2024a). See Appendix G for technical details.

**Results**. We identify independent linear subspaces for color and shape in the intermediate representations produced in the early layers of CLIP-pretrained ViTs (Figure 3b). In other words, we

---

[6]DAS has recently come under scrutiny for being too expressive (and thus not faithful to the model's internal algorithms) when misused (Makelov et al. (2023), cf. Wu et al. (2024b)). Following best practices, we deploy DAS on the residual stream. We use the `pyvene` library (Wu et al., 2024a) for all DAS experiments.

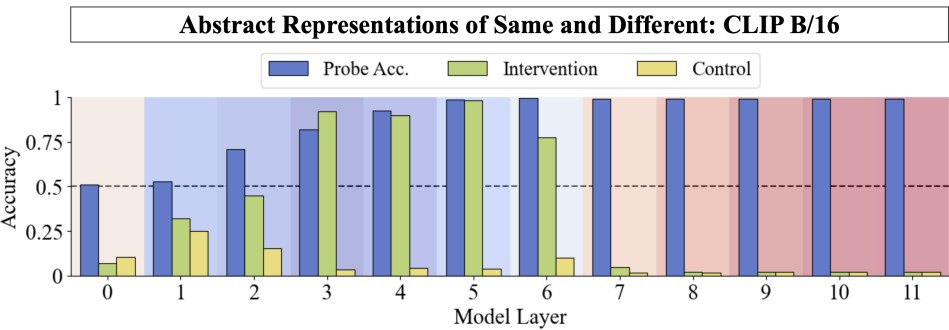

Figure 5: **Linear probing and intervention results**. We probe for the intermediate same-different judgments required to perform the RMTS task (blue). Probe performance reaches ceiling at around layer 5 and maintains throughout the rest of the model. We use the directions defined by the linear probe to intervene on model representations and flip an intermediate judgment (green). This intervention succeeds reliably at layer 5 but not deeper. We add a vector that is consistent with a pair's exhibited same-different relation as a control (yellow). This has little effect. The background is colored according to the heatmap in Figure 2c (blue=local heads; red=global heads).

can extract either color or shape information from one object and inject it into another object. This holds for both discrimination and RMTS tasks. One can conclude that at least one function of the perceptual stage is to form disentangled local object representations, which are then used to solve same-different tasks. Notably, these local object representations are formed in the first few layers and become increasingly irrelevant in deeper layers; intervening on the intermediate representations of object tokens at layers 5 and beyond results in chance intervention performance or worse. DINOv2-pretrained ViTs provide similar results (Appendix D), whereas other models exhibit these patterns less strongly (Appendix I). We present a control experiment in Appendix H, which further confirms our interpretation of these results.

## 5   The Relational Stage

We now characterize the relational stage, where tokens within one object largely attend to tokens in the other object. We hypothesize that this stage takes in the object representations formed in the perceptual stage and computes relational same-different operations over them. We find that the operations implemented by these relational layers are somewhat abstract in that 1) they do not rely on memorizing individual objects and 2) one can identify abstract *same* and *different* representations in the RMTS task, which are constant even as the perceptual qualities of the object pairs vary.

**Methods — Patching Novel Representations**.   In Section 4, we identify independent linear subspaces encoding shape and color. Does the *content* of these subspaces matter to the same-different computation? One can imagine an ideal same-different relation that is completely abstracted away from the particular properties of the objects being compared. In this setting, a model could accurately judge "same" vs. "different" for object representations where colors and shapes are represented by arbitrary vectors. To study this, we intervene on the linear subspaces for either shape or color for both objects in a pair, replacing the content found therein with novel representations (see Figure 4a). To create a "different" example, we start with a "same" image and replace the shape (or color) representations of both objects with two different novel representations; to create a "same" example, we start with a "different" image and replace them with two identical novel representations. We then assess whether the model's decision changes accordingly. We generate novel representations using four methods: 1) we *add* the representations found within the linear subspaces corresponding to two randomly sampled objects in an IID validation set, 2) we *interpolate* between these representations, 3) we *sample* each dimension randomly from a distribution of embeddings, and 4) we sample each dimension from an OOD *random* distribution (a $\mu = 0$ normal). See Appendix J for technical details.

**Results**.   The results of patching novel representations into a CLIP-pretrained ViT are presented in Figure 4b. Overall, we find the greatest success when patching in *added* representations, followed by *interpolated* representations. We observe limited success when patching *sampled* representations, and no success patching *random* vectors. All interventions perform best in layers 2 and 3, towards

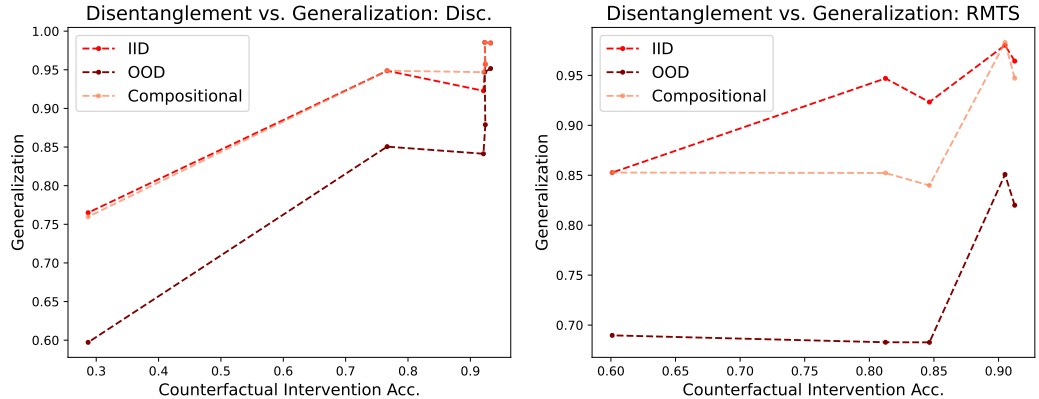

Figure 6: We average the best counterfactual intervention accuracy for shape and color and plot it against IID, OOD, and Compositional Test set performance for CLIP, DINO, DINOv2, ImageNet, MAE, and from-scratch B/16 models. We observe that increased disentanglement (i.e. higher counterfactual accuracy) correlates with downstream performance. The from-scratch model achieved only chance IID performance in RMTS, so we omitted it from the analysis.

the end of the perceptual stage. Overall, this points to a limited form of abstraction in the relational stage. The same-different operation is somewhat abstract—while it cannot operate over completely arbitrary vectors, it *can* generalize to additions and interpolations of shape & color representations, indicating that it does not rely on rote memorization of specific objects. Results for CLIP-pretrained ViTs on RMTS and other models on both tasks are found in Appendix K. Results for DINOv2 are found in Appendix D. Other models largely produce similar results to CLIP in this analysis.

**Methods — Linear Interventions**.    The RMTS task allows us to further characterize the relational stage, as it requires first forming then comparing intermediate representations of *same* and *different*. Are these intermediate representations abstract (i.e. invariant to the perceptual qualities of the object pairs that underlie them)? We linearly probe for intermediate same or different judgments from the collection of tokens corresponding to object pairs. The probe consists of a linear transformation mapping the residual stream to two dimensions representing *same* and *different*. Each row of this transformation can be viewed as a direction $d$ in the residual stream corresponding to the value being probed for (e.g. $d_{\text{same}}$ is the linear direction representing *same*). We train one probe for each layer on images from the model's train set and test on images from a test set. To understand whether the directions discovered by the probe are causally implicated in model behavior, we create a counterfactual intervention (Nanda et al., 2023). In order to change an intermediate judgment from *same* to *different*, we add the direction $d_{\text{diff}}$ to the intermediate representations of objects that exhibit the *same* relation. We then observe whether the model behaves as if this pair now exhibits the *different* relation.[7] We run this intervention on images from the model's test set. We also run a control intervention where we add the incorrect direction (e.g., we add $d_{\text{same}}$ when the object pair is already "same"). This control intervention should not reliably flip the model's downstream decisions.

**Results**.    Probing and linear intervention results for a CLIP-pretrained ViT are shown in Figure 5. We observe that linear probe performance peaks in the middle layers of the model (layer 5) and then remains high. However, our linear intervention accuracy peaks at layer 5 and then drops precipitously. Notably, layer 5 also corresponds to the peak of within-pair attention (see Figure 2c). This indicates that—at least in layer 5—there exists a single direction representing *same* and a single direction representing *different*. One can flip the intermediate same-different judgment by adding a vector in one of these directions to the residual streams of any pair of objects. Finally, the control intervention completely fails throughout all layers, as expected. Thus, CLIP ViT does in fact generate and operate over abstract representations of same and different in the RMTS task. We find similar results for a DINOv2 pretrained model (see Appendix D), but not for others (see Appendix K).

---

[7]Prior work normalizes the intervention directions and searches over a scaling parameter. In our setting, we find that simply adding the direction defined by probe weights without rescaling works well. See Appendix K for further exploration. We use `transformerlens` for this intervention (Nanda & Bloom, 2022).

# 6  Disentanglement Correlates with Generalization Performance

Object representations that disentangle perceptual properties may enable a model to generalize to out-of-distribution stimuli. Specifically, disentangled visual representations may enable compositional generalization to unseen combinations of perceptual properties (Higgins et al. (2018); Bengio et al. (2013), cf. Locatello et al. (2019))[8]. To investigate the relationship between disentanglement and generalization, we fine-tune CLIP, ImageNet, DINO, DINOv2, MAE, and randomly-initialized ViTs on a new dataset where each shape is only ever paired with two distinct colors. We then repeat our analyses in Section 4 to identify independent linear subspaces for shape and color.[9] We evaluate models in 3 settings: 1) on an IID test set consisting of observed shape-color combinations, 2) on a compositional generalization test set consisting of unobserved shape-color combinations (where each shape and each color have been individually observed), and 3) on an OOD test set consisting of completely novel shapes and colors. We plot the relationship between disentanglement (i.e. counterfactual intervention accuracy) and overall performance on these model evaluations. We find a consistent trend: more disentangled representations correlates with downstream model performance in all cases (See Figure 6).

# 7  Failure Modes

Previous sections have argued that pretrained ViTs that achieve high performance when finetuned on same-different tasks implement a two-stage processing pipeline. In this section, we argue that both perceptual and relational stages can serve as failure points for models, impeding their ability to solve same-different tasks. In practice, tasks that rely on relations between objects likely have perceptual and relational stages that are orders of magnitude more complex than those we study here. The results presented herein indicate that solutions targeting *either* the perceptual (Zeng et al., 2022) or relational (Bugliarello et al., 2023) stages may be insufficient for producing the robust, abstract computations that we desire.

**Perceptual and Relational Regularizers**.   We introduce two loss functions, designed to induce disentangled object representations and multi-stage relational processing, respectively. When employing the *disentanglement loss*, we introduce token-level probes that are optimized to predict shape information from one linear subspace (e.g., the first 384 dimensions) of the representations generated at an intermediate layer of the model and color information from the complementary linear subspace at that same layer (layer 3, in our experiments). These probes are optimized during training, and the probe loss is backpropagated through the model. This approach is motivated by classic work on disentangled representations (Eastwood & Williams, 2018). The *pipeline loss* is designed to encourage discrete, specific stages of processing by regularizing the attention maps to maximize the attention pattern scores defined in Section 3. Specifically, early layers are encouraged to maximize attention within-object, then within-pair, and finally (in the case of RMTS stimuli) between-pair. See Appendix L for technical details. Note that the disentanglement loss targets the perceptual stage of processing, whereas the pipeline loss targets both perceptual and relational stages.

**Results**.   First, we note that models trained from scratch on the discrimination task do not clearly distinguish between perceptual and relational stages (Figure 2b). Thus, we might expect that a model trained on a limited number of shape-color combinations would not learn a robust representation of the same-different relation. Indeed, Table 1 confirms this. However, we see that either the disentanglement loss or the pipeline loss is sufficient for learning a generalizable representation of this relation.

Similarly, we find that models trained from scratch on the RMTS task only achieve chance performance. However, in this case we must include *both* disentanglement and pipeline losses in order to induce a fairly general (though still far from perfect) hierarchical representation of same-different. This provides evidence that models may fail at either the perceptual *or* relational stages: they might

---

[8]Though Locatello et al. (2019) find that then-current measurements of of disentanglement fail to correlate with downstream performance in variational autoencoders, many of those models failed to produce disentangled representations in the first place.

[9]In order to train these interventions, we generate counterfactual images, some of which contain shape-color pairs that are not seen during model training. However, we emphasize that we only use these interventions to derive a disentanglement metric, not to train model weights.

| Task | Disent. Loss | Pipeline Loss | Train Acc. | Test Acc. | Comp. Acc. |
|------|:---:|:---:|---|---|---|
| Disc. | – | – | 77.3% | 76.5% | 76.0% |
| Disc. | ✓ | – | 97.3% (+20) | 94.6% (+18.1) | 86.1% (+10.1) |
| Disc. | – | ✓ | 95.6% (+18.3) | 93.9% (+17.4) | 92.3% (+16.4) |
| RMTS | – | – | 49.2% | 50.1% | 50.0% |
| RMTS | ✓ | – | 53.9% (+4.7) | 54.4% (+4.3) | 54.1% (+4.1) |
| RMTS | – | ✓ | 66.1% (+16.9) | 50.1% | 50.1% (+0.1) |
| RMTS | ✓ | ✓ | 95.1% (+45.9) | 94.1% (+44) | 77.4% (+27.4) |

Table 1: **Performance of ViTs trained from scratch with auxiliary losses**. Adding either a disentanglement loss term to encourage disentangled object representations (**Disent. Loss**) *or* a pipeline loss to encourage two-stage processing in the attention heads (**Pipeline Loss**) boosts test accuracy and compositional generalization (**Comp. Acc.**) for the discrimination task. *Both* auxiliary losses are required to boost accuracy for the RMTS task.

fail to produce the correct types of object representations, and/or they might fail to execute relational operations over them. See Appendix M for further analysis.

## 8   Discussion

**Related Work**.   This work takes inspiration from the field of mechanistic interpretability, which seeks to characterize the algorithms that neural networks implement (Olah, 2022). Though many of these ideas originated in the domain of NLP (Wang et al., 2022; Hanna et al., 2024; Feng & Steinhardt, 2023; Wu et al., 2024c; Merullo et al., 2023; Geva et al., 2022; Meng et al., 2022) and in toy settings (Nanda et al., 2022; Elhage et al., 2022; Li et al., 2022), they are beginning to find applications in computer vision (Fel et al., 2023; Vilas et al., 2024; Palit et al., 2023). These techniques augment an already-robust suite of tools that visualize the features (rather than algorithms) that vision models use (Olah et al., 2017; Selvaraju et al., 2017; Simonyan et al., 2014). Finally, this study contributes to a growing literature employing mechanistic interpretability to address debates within cognitive science (Millière & Buckner, 2024; Lepori et al., 2023a; Kallini et al., 2024; Traylor et al., 2024).

**Conclusion**.   The ability to compute abstract visual relations is a fundamental aspect of biological visual intelligence and a crucial stepping stone toward useful and robust artificial vision systems. In this work, we demonstrate that some fine-tuned ViTs adopt a two-stage processing pipeline to solve same-different tasks—despite having no obvious inductive biases towards this algorithm. First, models produce disentangled object representations in a perceptual stage; models then compute a somewhat abstract version of the same-different computation in a relational stage. Finally, we observe a correlation between disentanglement and generalization and note that models might fail to learn *either* the perceptual or relational operations necessary to solve a task.

Why do CLIP and DINOv2-pretrained ViTs perform favorably and adopt this two-stage algorithm so cleanly relative to other pretrained models? Raghu et al. (2021) find that models pretrained on more data tend to learn local attention patterns in early layers, followed by global patterns in later layers. Thus, pretraining scale (rather than training objective) might enable these models to first form local object representations, which are then used in global relational operations. Future work might focus on pinning down the precise relationship between data scale and relational reasoning ability, potentially by studying the training dynamics of these models. Additionally, future work might focus on characterizing the precise mechanisms (e.g. the attention heads and MLPs) used to implement the perceptual and relational stages, or generalize our findings to more complex relational tasks.

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

## A   Dataset Details

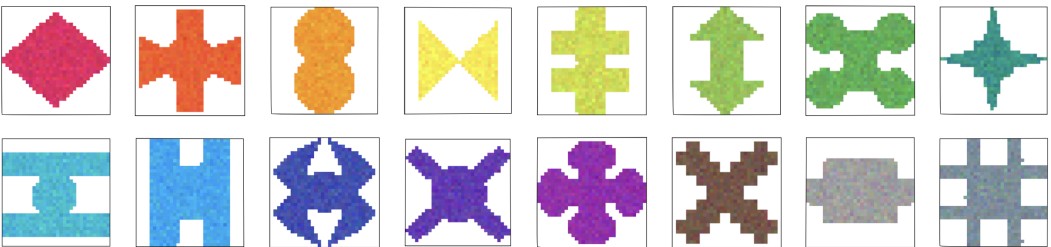

Figure 7: **All 16 unique shapes and colors used to construct the Discrimination and RMTS tasks**. There are thus $16 \times 16 = 256$ unique objects in our same-different datasets.

### A.1   Constructing the Objects

Figure 7 demonstrates a single instance of the 16 shapes and 16 colors used in our datasets. Any shape can be paired with any color to create one of 256 unique objects. Note that object colors are not uniform within a given object. Instead, each color is defined by three different Gaussian distributions—one for each RGB channel in the image—that determine the value of each object pixel. For example, the color red is created by these three distributions: $\mathcal{N}(\mu = 233, \sigma = 10)$ in the red channel, $\mathcal{N}(\mu = 30, \sigma = 10)$ in the green channel, and $\mathcal{N}(\mu = 90, \sigma = 10)$ in the blue channel. All color distributions have a variance fixed at 10 to give them an equal degree of noise. Any sampled values that lie outside of the valid RGB range of $[0, 255]$ are clipped to either 0 or 255. Object colors are re-randomized for every image, so no two objects have the same pixel values even if they are the same color. This was done to prevent the models from learning simple heuristics like comparing single pixels in each object.

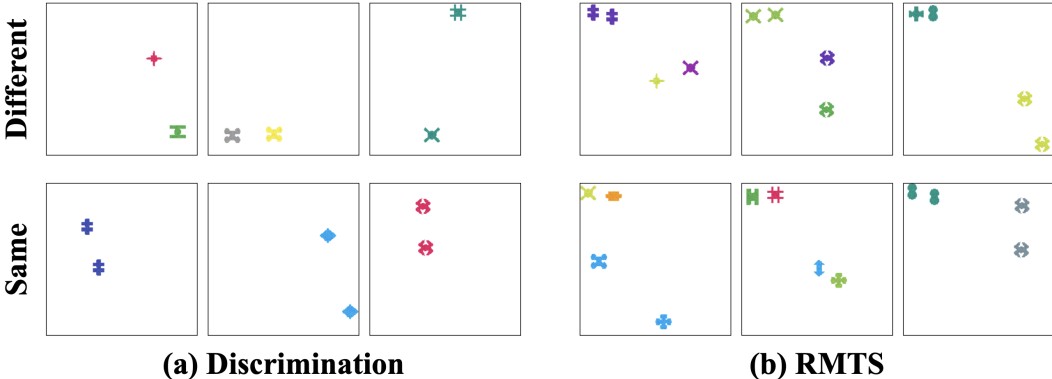

**(a) Discrimination**       **(b) RMTS**

Figure 8: **More examples of stimuli for the discrimination and RMTS tasks**. The top row shows "different" examples, while the bottom row shows "same" examples. Note that "different" pairs may differ in one or both dimensions (shape & color).

### A.2   Constructing the Datasets

The train, validation, and test sets for both the discrimination and RMTS tasks each contain $6,400$ unique stimuli: $3,200$ "same" and $3,200$ "different." To construct a given dataset, we first generate all possible same and different pairs of the 256 unique objects (see Figure 7). We consider two objects to be the same if they match in both shape and color—otherwise, they are different. Next, we randomly select a subset of the possible object pairs to create the stimuli such that each unique object is in at least one pair. For the RMTS dataset, we repeat this process to select same and different pairs of *pairs*.

Each object is resized (from $224 \times 244$ pixel masks of the object's shape) such that it is contained within a single ViT patch for B/32 models or four ViT patches for B/16 & B/14 models. For B/32 and

B/16 models, objects are roughly $28 \times 28$ pixels in size; for B/14 models (DINOv2 only), objects are roughly $21 \times 21$ pixels in size. These choices in size mean that a single object can be placed in the center of a $32 \times 32$ (or $28 \times 28$) pixel patch with a radius of 4 pixels of extra space around it. This extra space allows us to randomly jitter object positions within the ViT patches.

To create a stimulus, a pair of objects is placed over a $224 \times 224$ pixel white background in randomly selected, non-overlapping positions such that objects are aligned with ViT patches. For RMTS stimuli, the second "display" pair is always placed in the top left corner of the image. Each object's position (including the "display" objects for RMTS) is also randomly jittered within the ViT patches it occupies. We consider two objects in a specific placement as one unique stimulus—in other words, a given pair of objects may appear in multiple images but in different positions. All object pairs appear the same number of times to ensure that each unique object is equally represented.

See Figure 8 for some more examples of stimuli from each task.

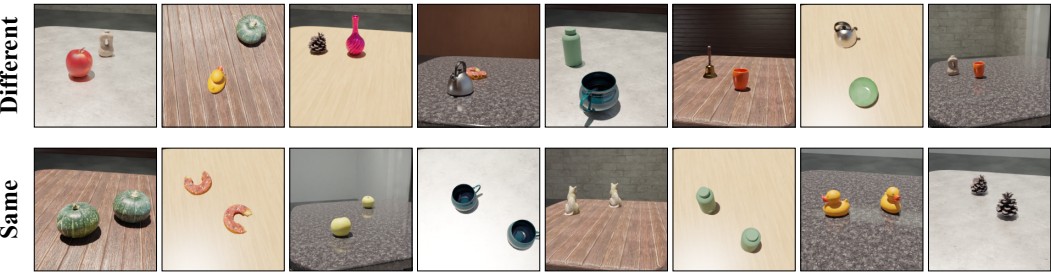

Figure 9: **Examples of stimuli from our photorealistic same-different evaluation dataset**. The top row contains "different" examples, while the bottom row contains "same" examples. Stimuli are constructed using 16 unique 3D models of objects placed on a table with a randomized texture; background textures are also randomized. Objects are randomly rotated and may be placed at different distances from the camera or occlude each other.

### A.3   Photorealistic Test Set

In order to ensure the robustness of the two-stage processing we observe in CLIP and DINOv2 on our artificial stimuli, we test models on a highly out-of-distribution photorealistic discrimination task. The test dataset consists of $1,024$ photorealistic same-different stimuli that we generated (see Figure 9). Each stimulus is a $224 \times 224$ pixel image depicting a pair of same or different 3D objects arranged on the surface of a table in a sunlit room. We created these images in Blender, a sophisticated 3D modeling tool, using a set of 16 unique 3D models of different objects that vary in shape, texture and color. To construct the dataset, we first generate all possible pairs of same or different objects, then select a subset of the possible "different" pairs such that each object appears in two pairs. This ensures that all objects are equally represented and that an equal number of "same" and "different" stimuli are created. We create 32 unique stimuli for each pair of objects by placing them on the table in eight random configurations within the view of four different camera angles, allowing partial occlusions. Each individual object is also randomly rotated around its $z$-axis in each image—because 11 of the objects lack rotational symmetry, these rotations provide an additional challenge, especially for "same" classifications.

We evaluate models that have been fine-tuned on the discrimination task from the main body of the paper (e.g. Figure 8a) in a zero-shot manner on the photorealistic dataset, meaning that there is no additional fine-tuning on the photorealistic dataset. We find that CLIP ViT attains a test accuracy of $93.9\%$ on the photorealistic dataset, while all other models attain chance level accuracy (e.g. DINOv2 attains an accuracy of $48\%$). We also find that CLIP performs two-stage processing on the photorealistic stimuli (see Figure 10a), and that the peaks in WO, WP, and BG attention all occur at the same exact layers as the artificial stimuli (i.e. in Figure 2). DINOv2 also displays similar two-stage processing despite its poor performance on the photorealistic task (see Figure 10b). Note that BG attention for both models is higher overall during the perceptual stage when processing the photorealistic stimuli compared to the artificial stimuli; this is likely because the photorealistic stimuli contain detailed backgrounds, while the backgrounds in the artificial stimuli are blank. Overall, these findings generalize our results from the toy setting presented in the main body of the paper.

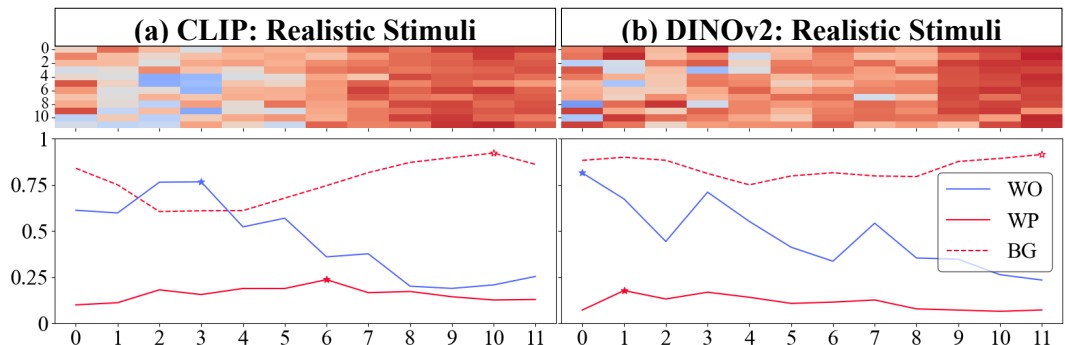

Figure 10: **Attention pattern analysis for CLIP and DINOv2 on the photorealistic discrimination task**. This figure follows the top row in Figure 2. **(a) CLIP**: As in Figure 2, WO peaks at layer 3, WP peaks at layer 6, and BG peaks at layer 10. BG attention is higher throughout the perceptual stage, leading to a lower perceptual score compared to the artificial discrimination task (i.e. fewer blue cells). **(b) DINOv2**: The attention pattern exhibits two stages, resembling the artificial setting (although the correspondence is somewhat looser than CLIP's, perhaps explaining DINOv2's poor zero-shot performance on the photorealistic task).

## B   ViT B/16: All Model Behavioral Results

See Tables 2, 3, 4, and 5 for behavioral results from all ViT-B/16 models trained on discrimination and RMTS tasks with either all 256 shape-color combinations or only 32 shape-color combinations. The "Pretraining Scale" column denotes the number of images (in millions) in a given model's pretraining dataset. The models are organized in descending order by pretraining scale. "Test Acc." refers to IID test accuracy. "Comp. Acc." refers to compositional generalization accuracy (for models trained on only 32 shape-color combinations). "Realistic Acc." (Table 2 only) refers to a model's zero-shot accuracy on the photorealistic evaluation dataset. CLIP and DINOv2—the two models with a pretraining scale on the order of 100 million images—attain near perfect test accuracy on the RMTS task. However, only CLIP attains high performance on the photorealistic dataset.

| Pretrain | Pretraining Scale ↓ | Train Acc. | Test Acc. | Realistic Acc. |
|----------|---------------------|------------|-----------|----------------|
| CLIP     | 400M                | 100%       | 99.3%     | 93.9%          |
| DINOv2   | 142M                | 100%       | 99.5%     | 48.0%          |
| ImageNet | 14.2M               | 100%       | 97.5%     | 53.0%          |
| DINO     | 1.28M               | 100%       | 95.6%     | 50.9%          |
| MAE      | 1.28M               | 100%       | 98.0%     | 52.4%          |
| –        | –                   | 95.9%      | 80.5%     | 49.9%          |

Table 2: **All behavioral results for ViT-B/16 models trained on all 256 shape-color combinations on the discrimination task**.

| Pretrain | Pretraining Scale ↓ | Train Acc. | Test Acc. | Comp. Acc. |
|----------|---------------------|------------|-----------|------------|
| CLIP     | 400M                | 98.1%      | 98.5%     | 98.5%      |
| DINOv2   | 142M                | 99.6%      | 98.5%     | 98.5%      |
| ImageNet | 14.2M               | 98.1%      | 95.7%     | 95.7%      |
| DINO     | 1.28M               | 98.1%      | 92.3%     | 94.7%      |
| MAE      | 1.28M               | 98.1%      | 94.9%     | 94.9%      |
| –        | –                   | 77.3%      | 76.5%     | 76.0%      |

Table 3: **All behavioral results for ViT-B/16 models trained on 32 shape-color combinations on the discrimination task**.

| Pretrain | Pretraining Scale ↓ | Train Acc. | Test Acc. |
|----------|---------------------|------------|-----------|
| CLIP | 400M | 100% | 98.3% |
| DINOv2 | 142M | 100% | 98.2% |
| ImageNet | 14.2M | 99.7% | 89.3% |
| DINO | 1.28M | 100% | 87.7% |
| MAE | 1.28M | 100% | 93.4% |
| – | – | 49.2% | 50.1% |

Table 4: **All behavioral results for ViT-B/16 models trained on all 256 shape-color combinations on the RMTS task**.

| Pretrain | Pretraining Scale ↓ | Train Acc. | Test Acc. | Comp. Acc. |
|----------|---------------------|------------|-----------|------------|
| CLIP | 400M | 100% | 98.0% | 98.3% |
| DINOv2 | 142M | 100% | 96.4% | 94.7% |
| ImageNet | 14.2M | 99.5% | 92.3% | 84.0% |
| DINO | 1.28M | 99.6% | 94.7% | 85.2% |
| MAE | 1.28M | 99.6% | 85.3% | 85.3% |
| – | – | 49.2% | 50.0% | 50.0% |

Table 5: **All behavioral results for ViT-B/16 models trained on 32 shape-color combinations on the RMTS task**.

# C    CLIP ViT-b32 Model Analyses

## C.1    Attention Pattern Analysis

See Figure 11.

## C.2    Perceptual Stage Analysis

See Figure 12.

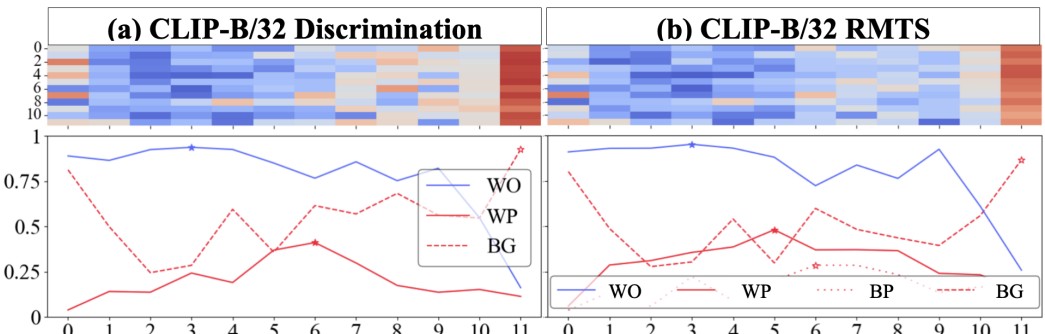

Figure 11: **CLIP B/32 attention pattern analysis**. See the caption of Figure 2 for figure and legend descriptions. The B/32 model follows the same stages of processing as CLIP ViT-B/16, and WO & WP peak at the same layers (3 and 6 for discrimination respectively; 3 and 5 for RMTS respectively). However, WO attention remains high for longer than B/16 models.

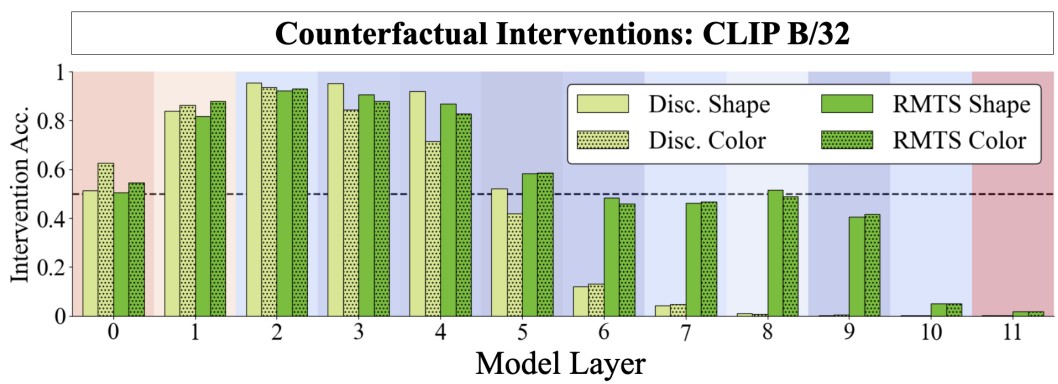

Figure 12: **CLIP B/32 DAS results**.

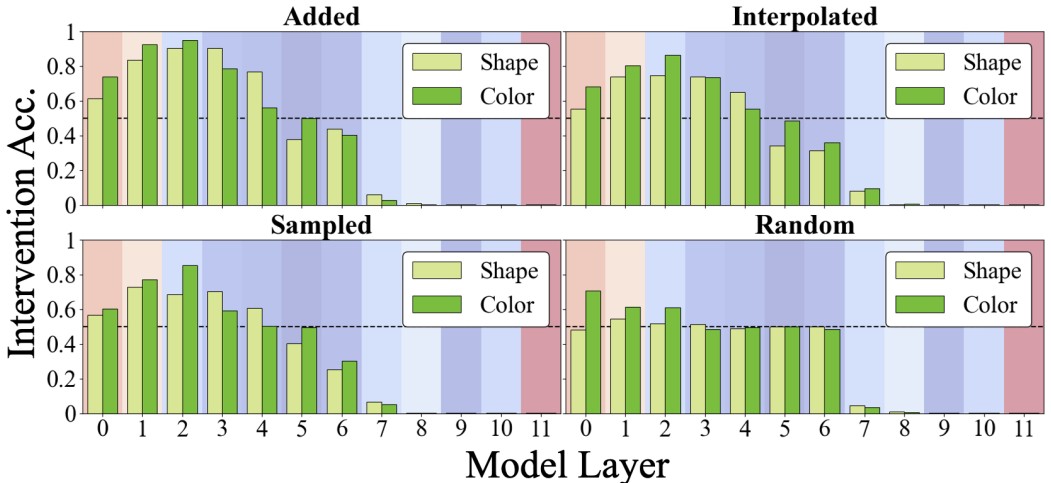

Figure 13: **CLIP B/32 relational stage analysis: Novel Representations**.

### C.3  Relational Stage Analysis

See Figure 13 for novel representations analysis. See Figure 14 for linear intervention analysis. We find broadly similar results as CLIP B/16.

### C.4  Generalization Results

We present CLIP-B/32 model results for models finetuned on all shape-color combinations, as well as only 32 shape-color combinations, as in Section 6. We present compositional generalization accuracy

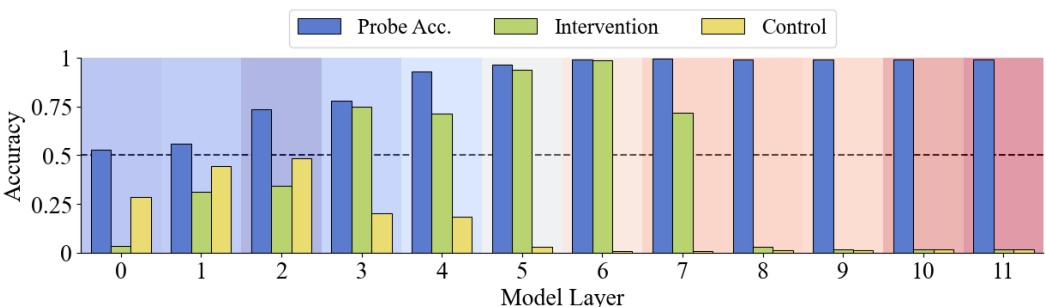

Figure 14: **CLIP B/32 relational stage analysis: Linear Intervention**.

| Task | # Combinations Seen | Train Acc. | IID Test Acc. | Comp. Gen. Acc. | OOD Acc. |
|------|---------------------|------------|---------------|-----------------|----------|
| Disc. | All | 100% | 99.6% | N/A | 95.8% |
| Disc. | 32 | 99.7% | 98.5% | 98.5% | 98.0% |
| RMTS | All | 100% | 97.4% | N/A | 90.5% |
| RMTS | 32 | 100% | 98.3% | 98.3% | 86.2% |

Table 6: **All behavioral results for CLIP-B/32 models**.

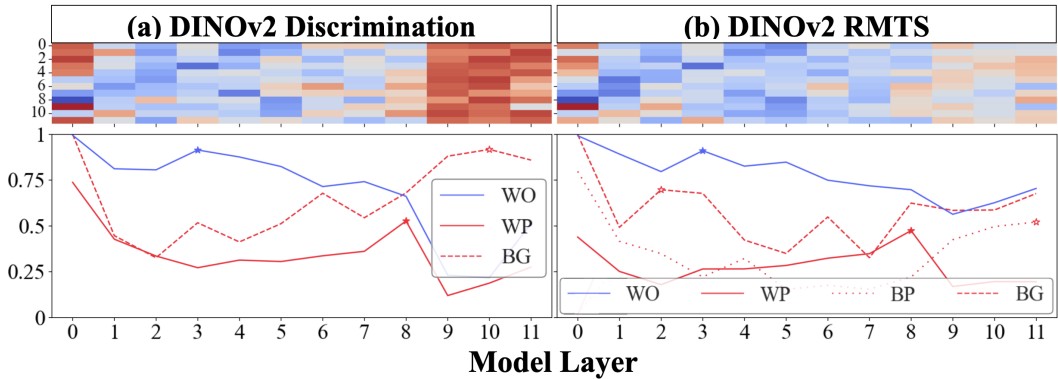

Figure 15: **DINOv2 attention pattern analysis**. See the caption of Figure 2 for figure and legend descriptions. Note that the stars in the line charts are placed differently in this figure compared to other attention pattern analysis figures. Instead of marking the maximal values of each type of attention across all 12 layers, the stars mark the maximal value excluding the 0th layer. This is because all types of attention spike in DINOv2 in the 0th layer.

(when applicable) as well as OOD generalization accuracy. We find that all models perform quite well in-distribution *and* under compositional generalization. Accuracy drops somewhat for RMTS OOD stimuli. All results are in presented in Table 6.

# D    DINOv2 Analyses

## D.1    Attention Map Analysis

See Figure 15 for DINOv2 attention pattern analyses. Like CLIP, DINOv2 displays two-stage processing (albeit somewhat less cleanly). One notable difference compared to CLIP is that all types of attention (WO, WP, BP, and BG) spike in the 0th layer. This might be related to DINOv2's positional encodings. Since the model was pretrained on images with a size of $518 \times 518$ pixels, the model's positional encodings are interpolated to process our $224 \times 224$ stimuli; this might cause an artifact in the attention patterns in the very beginning of the model. Disregarding this spike, the stages of processing follow CLIP. In the discrimination task (Figure 15a), within-object attention peaks at layer 3 (disregarding the initial peak), followed by within-pair and finally background attention. In the RMTS task (Figure 15b), within-object attention peaks at layer 3, followed by within-pair attention at layer 8, and finally between-pair attention in the final layer. Background attention remains relatively high throughout the model, indicating that DINOv2 might make greater use of register tokens to solve the RMTS task compared to other models.

## D.2    Perceptual Stage Analysis

See Figure 16 for perceptual stage analysis of DINOV2-pretrained ViTs. Overall, we find highly disentangled object representations in these models.

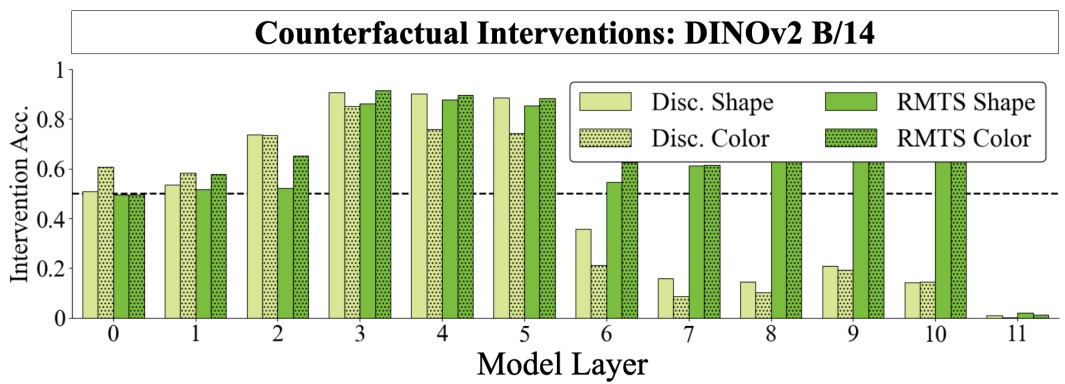

Figure 16: **DAS results for DINOv2 ViT-B/14**.

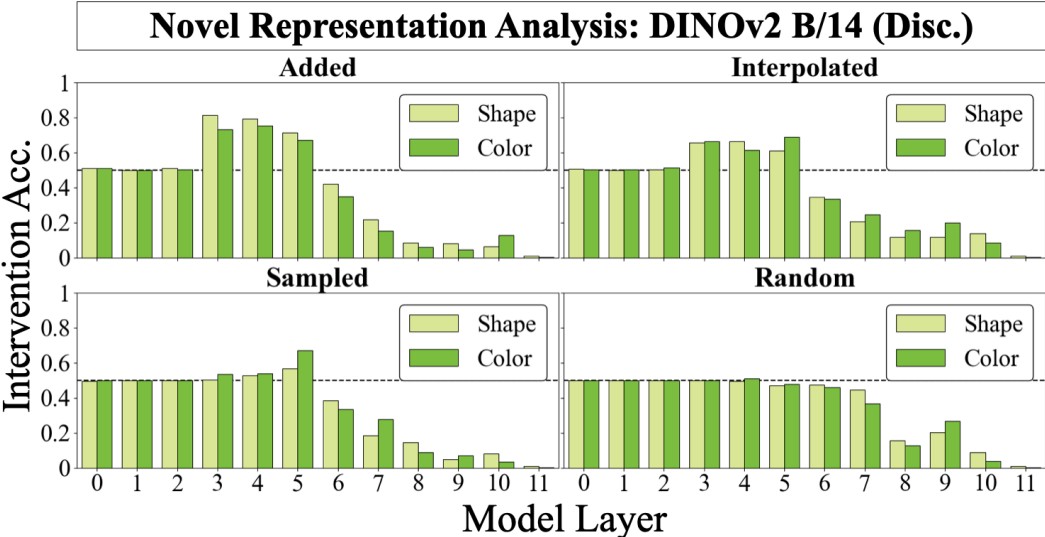

Figure 17: **Novel Representation Analysis for DINOv2 ViT-B/14 (Disc.)**.

### D.3 Relational Stage Analysis

#### D.3.1 Novel Representation Analysis

See Figure 17 and 18 for novel representation analysis of DINOV2-pretrained ViTs for the discrimination and RMTS tasks. These results replicate those found using CLIP-pretrained ViTs.

#### D.3.2 Abstract Representations of *Same* and *Different*

See Figure 19 for linear probing and intervention results for DINOV2-pretrained ViTs. We find that the intervention works extremely well for these models, replicating our results on CLIP-pretrained ViTs.

## E Attention Pattern Analyses

See Figure 20 for attention pattern analyses for ImageNet, DINO, and MAE ViT on the Discrimination and RMTS tasks. ImageNet loosely demonstrates two-stage processing like CLIP and DINOv2. On the other hand, DINO and MAE do not display two stage processing; instead, local and global processing appears to be mixed throughout the models. DINO and MAE also receive the smallest scale pretraining compared to the other models (see the Pretraining Scale column in Table 2); this provides further support for our intuition that pretraining scale results in two-stage processing.

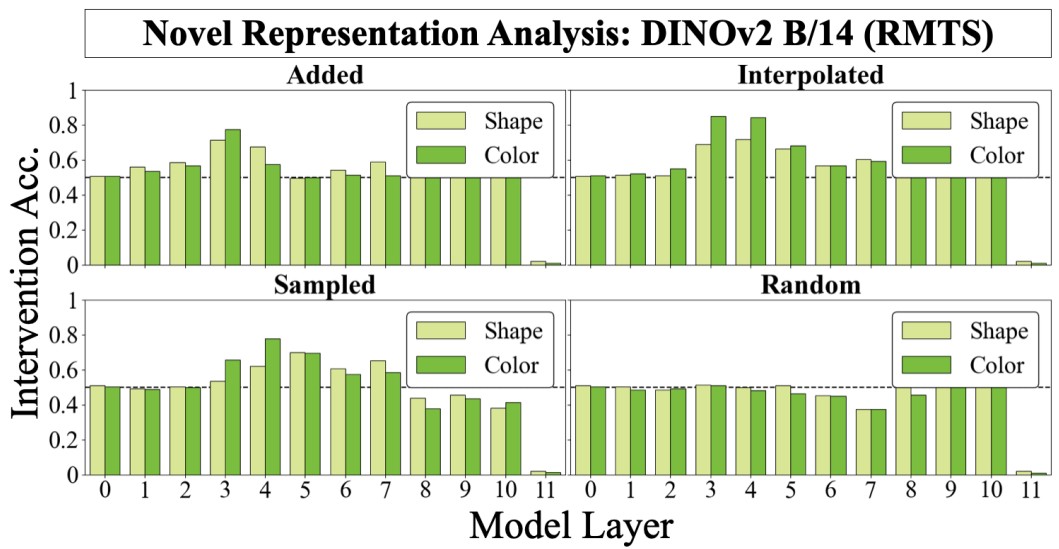

Figure 18: **Novel Representation Analysis for DINOv2 ViT-B/14 (RMTS)**.

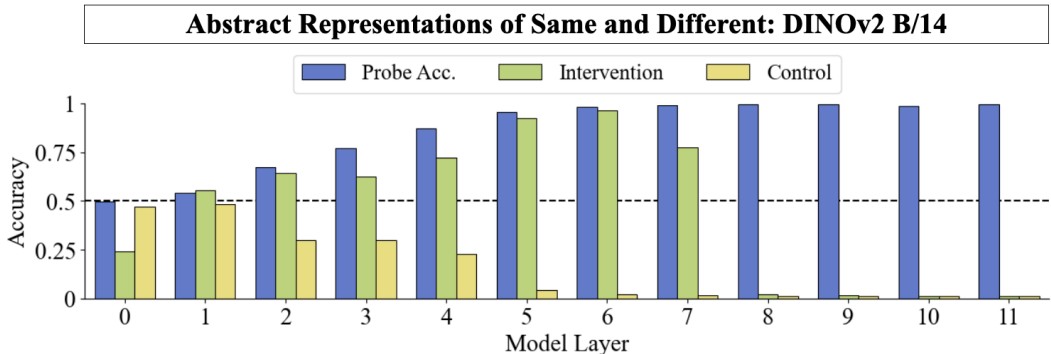

Figure 19: **Linear probe & intervention analysis for DINOv2 ViT-B/14**.

## F   Two Internal Algorithms Examined in Greater Detail

While the attention head analysis in Section 3 shows that different models use qualitatively different internal algorithms to solve same-different tasks, the specific computations involved in these algorithms are less clear. What exactly is happening during CLIP's perceptual stage, for example? In this section, we seek to build an intuitive picture of the algorithms learned by two models on the discrimination task: CLIP ViT-B/16, and a randomly initialized ViT-B/16 (From Scratch).

To do this, we examine the attention patterns produced by individual attention heads throughout each model. Figure 21 displays attention patterns extracted from four randomly selected individual attention heads (black & white heatmaps) in response to the input image on the left. For CLIP, the examined heads are: layer 1, head 5 (local head); layer 5, head 9 (local head); layer 6, head 11 (global head); and layer 10, head 6 (global head). For the from scratch model, the heads are: layer 1, head 8; layer 5, head 11; layer 6, head 3; and layer 10, head 8. For visualization purposes, the attention patterns are truncated to include only the indices of the two objects' tokens; since each object occupies four ViT patches, this results in an $8 \times 8$ grid for each attention head. The `src` axis ($y$-axis) in Figure 21 indicates the source token, while the `dest` axis ($x$-axis) indicates the destination token (attention flows from `src`$\longrightarrow$`dest`). The actual tokens in the input image are also visualized along these axes.

Based on these attention patterns, we visualize how CLIP processes an image to solve the discrimination task in Figure 22. **1. Embedding**: The model first tokenizes the image and embeds these

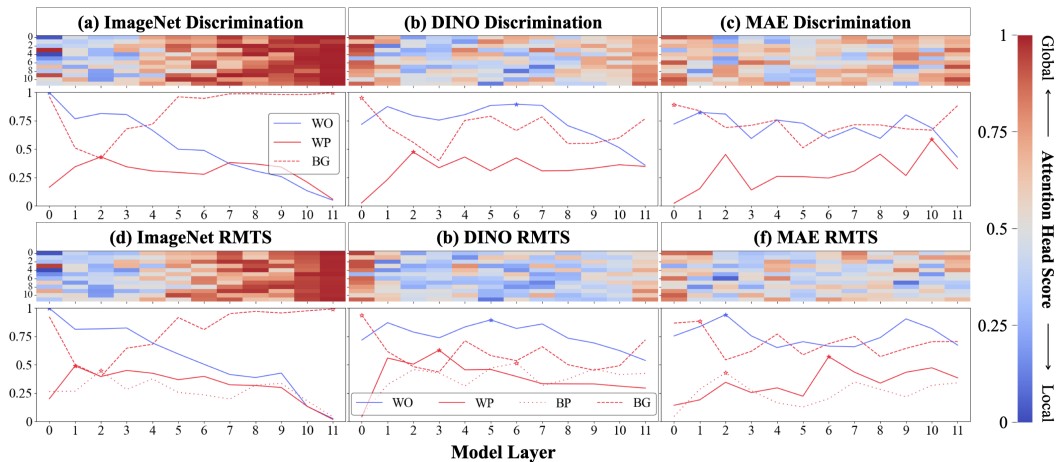

Figure 20: **ImageNet, DINO, and MAE ViT attention pattern analysis**. See the caption of Figure 2 for figure and legend descriptions. Like CLIP and DINOv2, ImageNet ViT displays two-stage processing on both the discrimination and RMTS tasks; however, performance of this model lags behind CLIP and DINOv2, possibly due to smaller pretraining scale (see the Pretraining Scale column in Table 2). DINO and MAE do not display two-stage processing. These two models are also pretrained on the smallest amount of data, further supporting our intuition that pretraining scale rather than objective results in two-stage processing.

tokens. Each object occupies four ViT-B/16 patches, so the objects are divided up into four tokens each. **2. Layer 1, Head 5**: During the early perceptual stage, the local attention heads appear to aid in the formation of object representations by performing low-level comparisons within objects. For example, head 5 in layer 1 compares object tokens from left to right within each object. Other attention heads in this layer perform such comparisons in other directions, such as right to left or top to bottom. **3. Layer 5, Head 9**: Towards the end of the perceptual stage, all object tokens within a single object attend to all other tokens within the same object. The four object tokens comprising each object have been pushed together in the latent space, and the model now "sees" a single object as a whole. **4. Layer 6, Head 11**: The model switches from predominantly local to predominantly global attention in this layer, and within-pair (WP) attention peaks. The whole-object representations formed during the perceptual stage now attend to each other, indicating that the model is comparing them. **5. Layer 10, Head 6**: The model appears to utilize object tokens (and background tokens) to store information, possibly the classification decision).

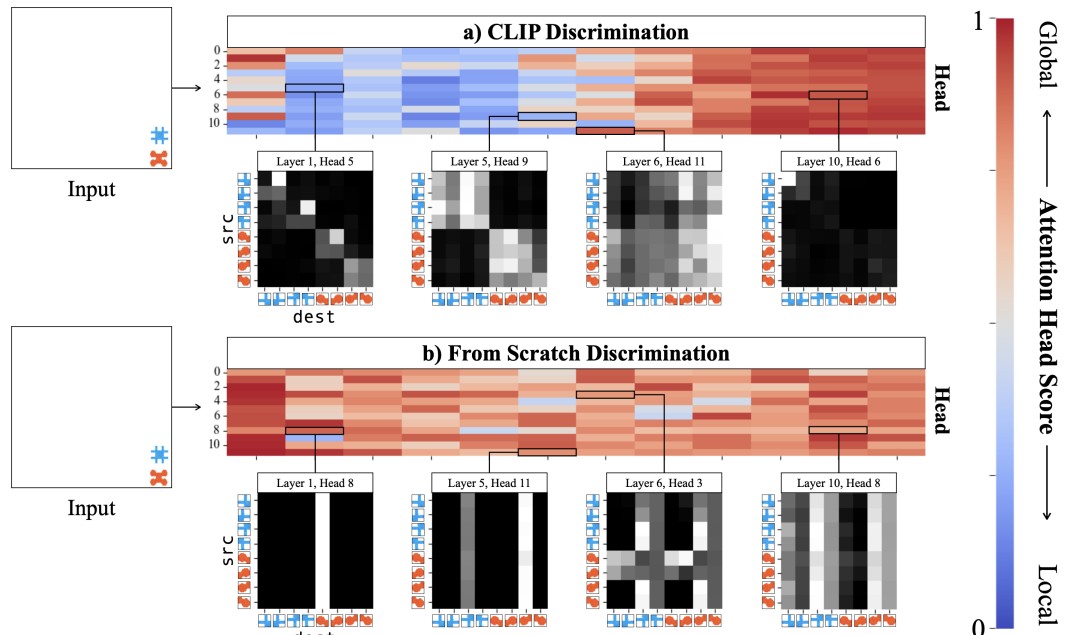

Figure 21: **Example attention head patterns for models trained on the discrimination task**. **(a) CLIP ViT-B/16**: On the left is an example input image, which is fed into the model. The heatmap is the same as Figure 2a—the $x$ and $y$-axes denote model layer and head index respectively, and the colors indicate the type of attention head as defined in Section 3 (local heads=blue, global heads=red). The attention patterns of four attention heads for this input image are displayed in black & white heatmaps below; white indicates higher attention values. The `src` axis indicates the source token, which is visually marked—recall that each object occupies four tokens each. The `dest` axis indicates the destination token. Individual objects attend to themselves during the perceptual stage (layers 0-5); objects begin to attend to the other object during the relational stage (layer 6 onwards). **(b) From Scratch ViT-B/16**: The analysis in (a) is repeated for a from scratch model trained on discrimination. The attention patterns are less interpretable throughout.

## G  Distributed Alignment Search Technical Details

**Approach**  We apply a form of distributed alignment search (Geiger et al., 2024) in order to assess whether the object representations formed by the perceptual stage of ViTs are disentangled with respect to shape and color (the two axes of variation present in our dataset). For ViT B/32 models, each object is contained within the bounds of a single patch, making DAS straightforward to run: we train an intervention over model representations corresponding to the patches containing individual objects that we wish to use as source and base tokens. For ViT B/16 models, each object is contained in four patches. Here, we train a single intervention that is shared between all four patches comprising the base and source objects. Importantly, because we wish to isolate *whole object representations*, rather than *patch representations*, we randomly shuffle the 4 patches comprising the source object before patching information into the base object. For example, the top-right patch of the base object might be injected with information from the bottom-left patch of the source object. This intervention should only succeed if the model contains a disentangled representation of the whole object, and if this representation is present in all four patches comprising that object. Given these stringent conditions, it is all the more surprising that DAS succeeds. During test, we intervene in a patch-aligned manner: The vector patched into the top-right corner of the base image representation is extracted from the top-right corner of the source image.

**Data**  To train the DAS intervention, we must generate counterfactual datasets for every subspace that we wish to isolate. To generate a discrimination dataset that will be used to identify a color subspace, for example, we find examples in the model's training set where objects only differ along the color dimension (e.g., `object₁` expresses `color₁`, `object₂` expressed `color₂`. We randomly

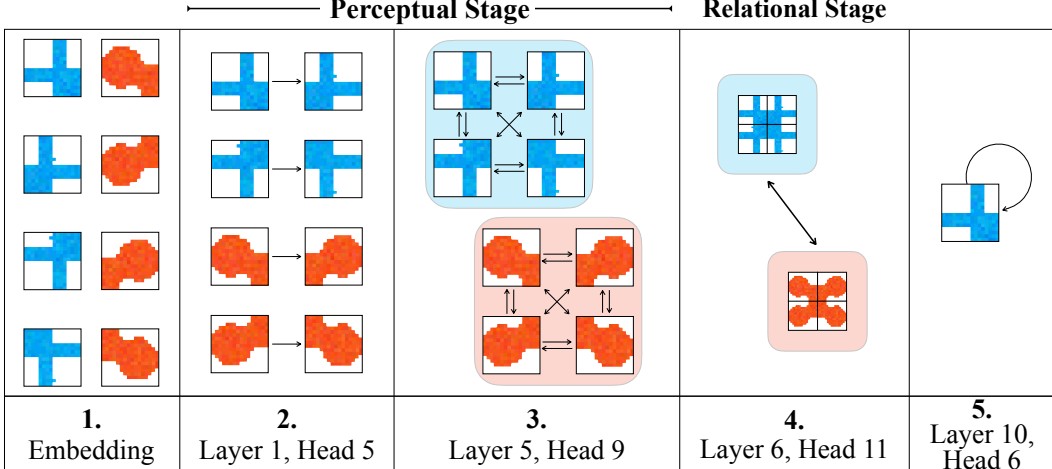

| **1.** | **2.** | **3.** | **4.** | **5.** |
|---|---|---|---|---|
| Embedding | Layer 1, Head 5 | Layer 5, Head 9 | Layer 6, Head 11 | Layer 10, Head 6 |

Figure 22: **How CLIP ViT-B/16 processes an example from the discrimination task**. Four attention heads are randomly selected from different stages in CLIP and analyzed on a single input image (see Figure 21). **1. Embedding**: The model first tokenizes the input image. Each object occupies four ViT patches. **2. Layer 1, Head 5**: During the perceptual stage, the model first performs low-level visual operations between tokens of individual objects. This particular attention head performs left-to-right attention within objects. **3. Layer 5, Head 9**: Near the end of the perceptual stage, whole-object representations have been formed. **4. Layer 6, Head 11**: During the relational stage, the whole-object representations are compared. **5. Layer 10, Head 6**: Object and background tokens are used as registers to store information—presumably the classification.

select one object to intervene on and generate a counterfactual image. WLOG consider intervening on $object_1$. Our counterfactual image contains one object (the counterfactual object) that expresses $color_2$. Our intervention is optimized to extract color information from the counterfactual object and inject it into $object_1$, changing the model's overall discrimination judgment from *different* to *same*. Importantly, the counterfactual image label is also *different*. Thus, our intervention is designed to work *only* if the intervention transfers color information. We follow a similar procedure for to generate counterfactuals that can be used to turn a *same* image into a *different* image. In this case, both base and counterfactual images are labelled *same*, but the counterfactual *same* image contains objects that are a different color than those in the base image. The counterfactual color is patched into one of the objects in the base image, rendering the objects in the base image *different* along the color axis.

For the RMTS DAS dataset, we generate counterfactuals similarly to the discrimination dataset. We select a pair of objects randomly (except *either* the display pair or sample pair). We then choose the source object in the other pair. We edit the color or shape property of just this source object, and use this as the counterfactual. For these datasets, the data is balanced such that 50% of overall labels are changed from *same* to *different*, but also 50% of intermediate pair labels are changed from *same* to *different*. Note that flipping one intermediate label necessarily flips the hierarchical label. Thus, if the source object is in a pair expressing the *same* relationship, then the counterfactual image will have the opposite label as the base image before intervention. In these cases, the intervention could succeed by transferring the hierarchical image label, rather than by transferring particular color or shape properties from one object to another. However, this only occurs approximately 50% of the time. That is because it occurs in 100% of samples when both pairs exhibit *same*, which occurs 25% of the time (half of the hierarchical *same* images), and 50% of the time when one pair exhibits *same* and the other exhibits *different*, which occurs 50% of the time (all of the hierarchical different images). However, this strategy provides exactly the incorrect incorrect result in the other 50% of cases. Nonetheless, this behavior might explain why RMTS DAS results maintain at around 50% deeper into the model.

For all datasets, we generate train counterfactual pairs from the model train split, validation pairs from the validation split, and test pairs from the model test split. We generate 2,000 counterfactual pairs for both splits. Note that in the case of models trained in the compositional generalization

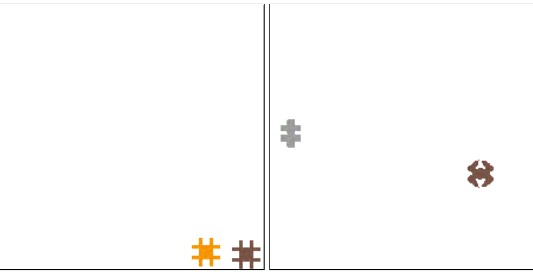

(a) Discrimination: Color counterfactual pair. The brown color from the object on the right will be patched into the orange color in the object on the left.

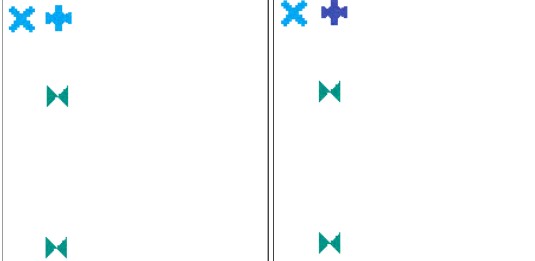

(b) RMTS: Color counterfactual pair. The dark blue color from the display pair object on the right will be patched into one of the green sample objects on the left.

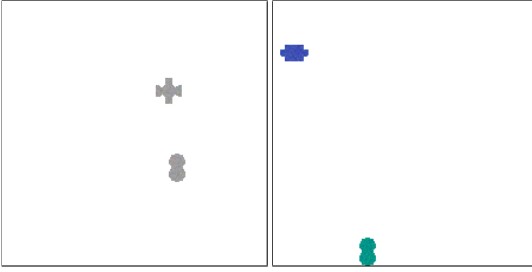

(c) Discrimination: Shape counterfactual pair. The two-circles shape from the object on the right will be patched into the cross shape in the object on the left.

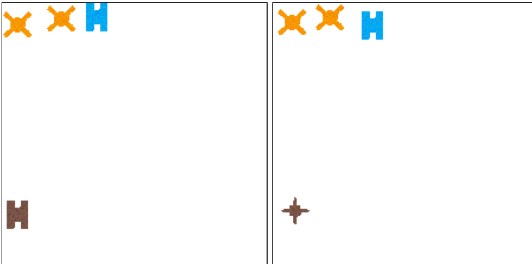

(d) RMTS: Shape counterfactual pair. The thin star shape from the sample pair object on the right will be patched into one of the orange objects on the left.

Figure 23: Counterfactual pairs used to train DAS interventions.

experiments (i.e. those found in Section 6), the counterfactual image may contain shape-color pairs that were not observed during training. However, training our interventions has no bearing on the model's downstream performance on held-out data, though correlation between disentanglement and compositional generalization is thus not extremely surprising. See Figure 23 for examples of counterfactual pairs used to train interventions.

**Intervention Details**   DAS requires optimizing 1) a rotation matrix over representations and 2) some means of identifying appropriate dimensions over which to intervene (Geiger et al., 2024). Prior work has largely heuristically selected *contiguous* subspaces over which to intervene (Geiger et al.,

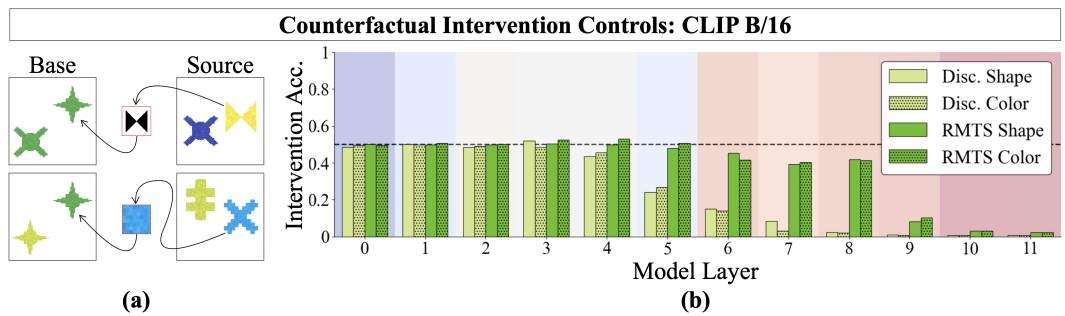

Figure 24: **Controls for DAS Analysis (Section 4)**. **(a)** We intervene on objects using the irrelevant source token in the counterfactual stimuli. For example, if the DAS interventions are meaningful, patching a bowtie shape into the base image in the top row should not change the model's decision to "same." **(b)** The controls fail to flip CLIP ViT's decisions at a rate above chance accuracy, indicating that the DAS results presented in Section 4 are indeed the result of meaningful interventions.

2024; Wu et al., 2024c). In this work, we relax this heuristic, identifying dimensions by optimizing a binary mask over model representations as we optimize the rotation matrix (Wu et al., 2024a). We follow best practices from differentiable pruning methods like continuous sparsification (Savarese et al., 2020), annealing a sigmoid mask into a binary mask over the course of training, using an exponential temperature scheduler. We also introduce an $L_0$ penalty to encourage sparse masking. We use default parameters suggested by the `pyvene` library for Boundless DAS, another DAS method that optimizes the dimensions over which to intervne. Our rotation matrix learning rate is 0.001, our mask learning rate is 0.01, and we train for 20 epochs for each subspace, independently for each model layer. We add a scalar multiplier of 0.001 to our $L_0$ loss term, which balances the magnitude of $L_0$ loss with the normal cross entropy loss that we are computing to optimize the intervention. Our temperature is annealed down to 0.005 over the course of training, and then snapped to binary during testing. Finally, we optimize our interventions using the Adam optimizer (Kingma & Ba, 2014). These parameters reflect standard practice for differentiable masking for interpretability (Lepori et al., 2023b).

## H    Perceptual Stage Analysis Controls

As a control for our DAS analysis presented in Section 4, we attempt to intervene using the incorrect source token in the counterfactual image. If this intervention fails, then it provides evidence that the information transferred in the standard DAS experiment is actually indicative of disentangled local object representations, rather than information that may be distributed across all objects. We note that this control could succeed at flipping *same* judgments to *different*, but will completely fail in the opposite direction. As shown in Figure 24, these controls do reliably fail to achieve above-chance counterfactual intervention accuracy.

## I    Perceptual Stage Analysis: Other Models

See Figures 25, 26, 27, and 28 for DINO, ImageNet, MAE, and From Scratch DAS results. We see that models broadly exhibit less disentanglement than CLIP and DINOv2.

## J    Novel Representations Analysis Technical Details

During the novel representations analysis of Section 5, we patch in novel vectors into the subspaces identified using DAS. Notably, we must patch into the {color, shape} subspace for *both* objects that we wish to intervene on, rather than just one. This is because we want to analyze whether the same-different relation can generalize to novel representations. For example, if two objects in a discrimination example share the same shape, but one is blue and one is red , we would like to know whether we can intervene to make the color property of each object an identical, novel vector, such

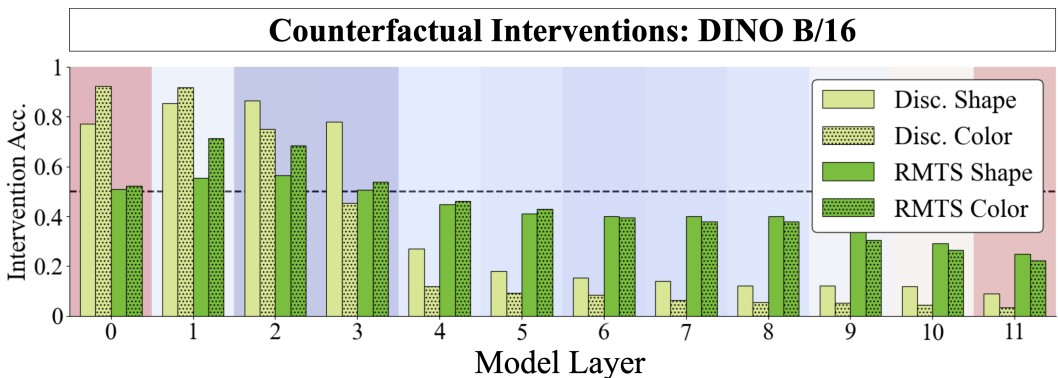

Figure 25: **DAS results for DINO ViT-B/16**.

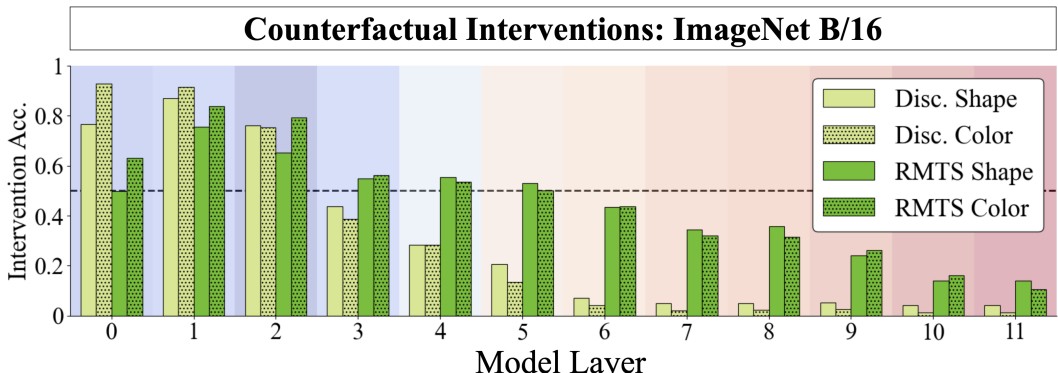

Figure 26: **DAS results for ImageNet ViT-B/16**.

that the model's decision will flip from *different* to *same*. We run this analysis on the IID test set of the DAS data.

We create these vectors using four different methods. For these methods, we first save the embeddings found in the subspaces identified by DAS for all images in the DAS Validation set.

1. **Addition**: We sample two objects in the validation set, and add their subspace embeddings. For ViT-B/16, we patch the resulting vector in a patch-aligned manner: The vector patched into the top-right corner of the base image representation is generated by adding the top-right corners of each embedding found within the subspace of the sampled validation images.

2. **Interpolation**: Same as Method 1, except vectors are averaged dimension-wise.

3. **Sampled**: We form one Gaussian distribution per embedding dimension using our saved validation set embeddings. We independently sample from these distributions to generate a vector that is patched into the base image. This single vector is patched into all four object patches for ViT-B/16.

4. **Random Gaussian**: We randomly sample from a normal distribution with mean 0 and standard deviation 1 and patch that into the base image. This single vector is patched into all four object patches for ViT-B/16.

## K  Relational Stage Analysis: Further Results

### K.1  CLIP B/16 RMTS Novel Representation Analysis

See Figure 29 for novel representation analysis on CLIP B/16, finetuned for the relational match to sample task.

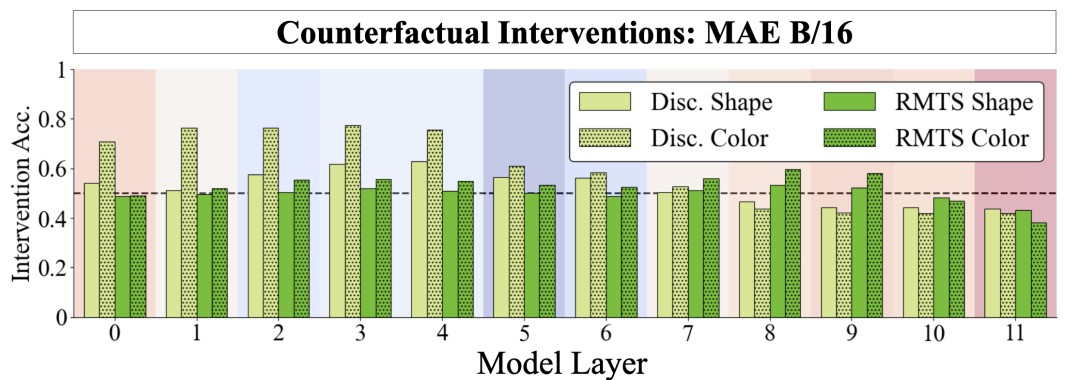

Figure 27: **DAS results for MAE ViT-B/16**.

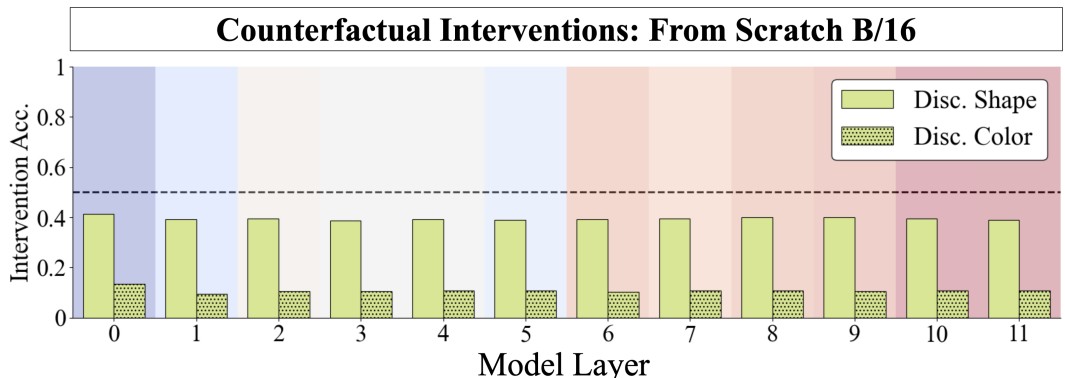

Figure 28: **DAS results for From Scratch ViT-B/16**.

### K.2  Novel Representation Analysis: Other Models

See Figures 30, 31, 32, 33, 34, 35, 36 for DINO Discrimination/RMTS, ImageNet Discrimination/RMTS, MAE Discrimination/RMTS and From Scratch Discrimination Novel Representation Analysis results.

### K.3  Abstract Representations of *Same* and *Different*

We run the linear probe and linear intervention analysis from Section 5 on DINO B/16, ImageNet B/16, and MAE B/16. We find that the intervention works much less well on these models than on DINOv2 B/16, CLIP B/16 or CLIP B/32. This indicates that these models are not using one abstract representation of *same* and one representation of *different* that is agnostic to perceptual qualities of the input image.

Additionally, we try to scale the directions that we are adding to the intermediate representations by 0.5 and 2.0 for DINO and ImageNet pretrained models, and find that neither of these versions of the intervention work much better well for either model. See Figure 37 and 38 for DINO and ImageNet results. See Figure 39 for MAE model results.

## L  Pipeline Loss Technical Details

To instill two-stage processing in ViTs trained from scratch on discrimination, we shape the model's attention patterns in different ways at different layers. In particular, within-object attention is encouraged in layers 3, 4, and 5, while between-object attention is encouraged in layers 6 and 7 (roughly following CLIP's two-stage processing; see Figure 2a). For models trained on RMTS, we additionally encourage between-pair attention in layers 8 and 9 (see Figure 2c). Finally, whenever we add the disentanglement loss, it is computed in layer 3 only.

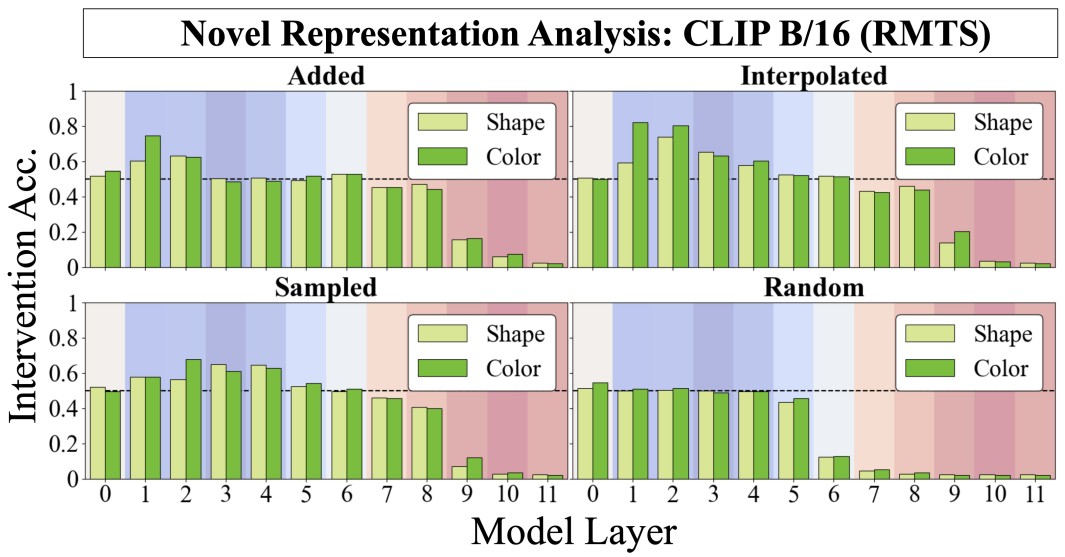

Figure 29: **Novel Representation Analysis for CLIP ViT-B/16 (RMTS)**.

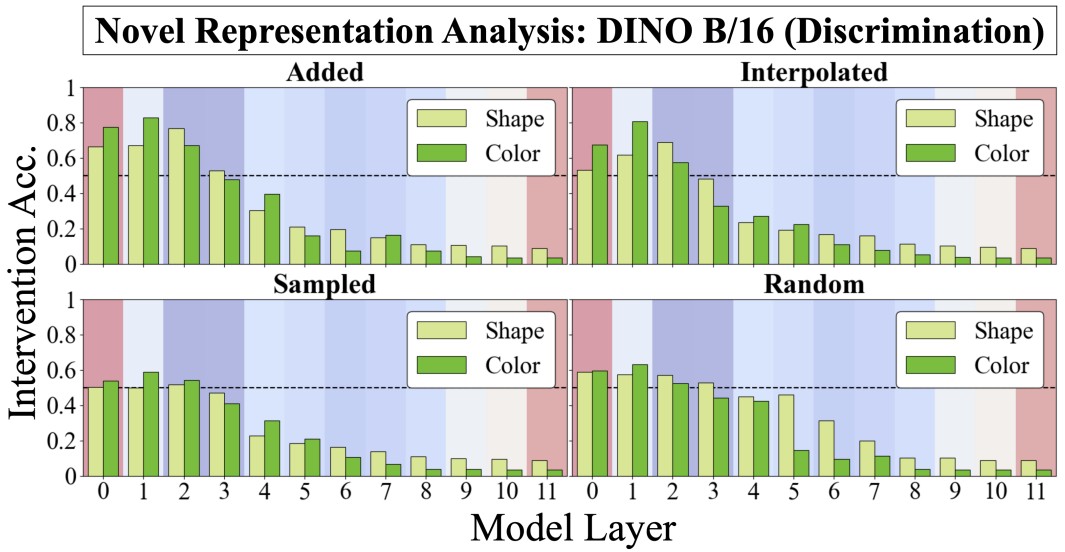

Figure 30: **Novel Representation Analysis for DINO ViT-B/16 (Disc.)**.

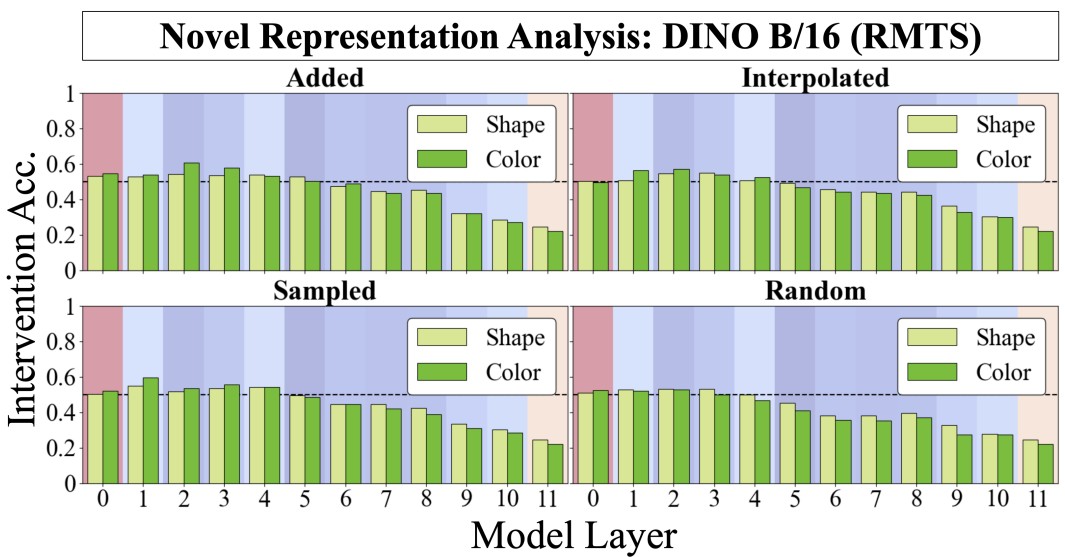

Figure 31: **Novel Representation Analysis for DINO ViT-B/16 (RMTS)**.

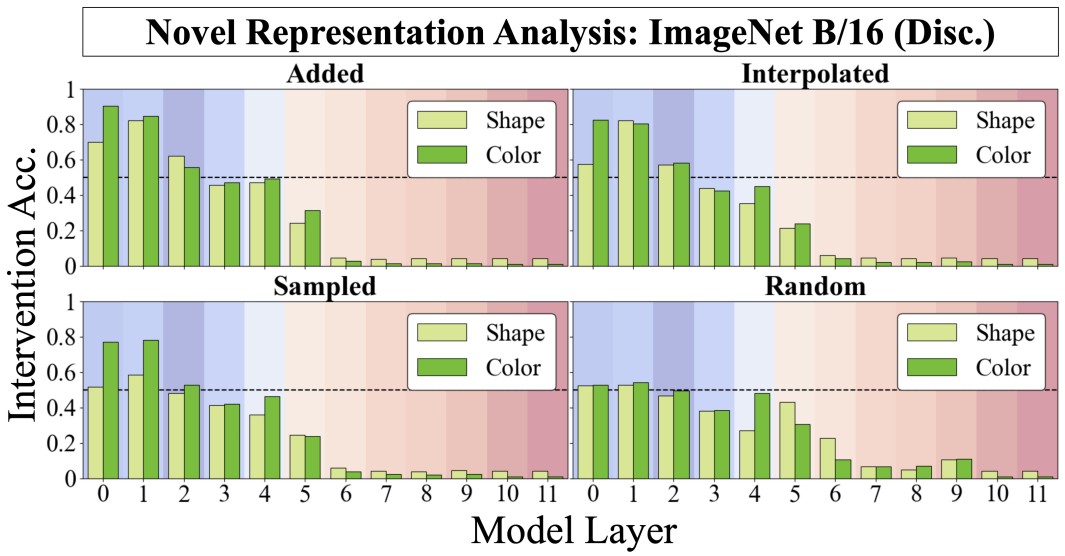

Figure 32: **Novel Representation Analysis for ImageNet ViT-B/16 (Disc.)**.

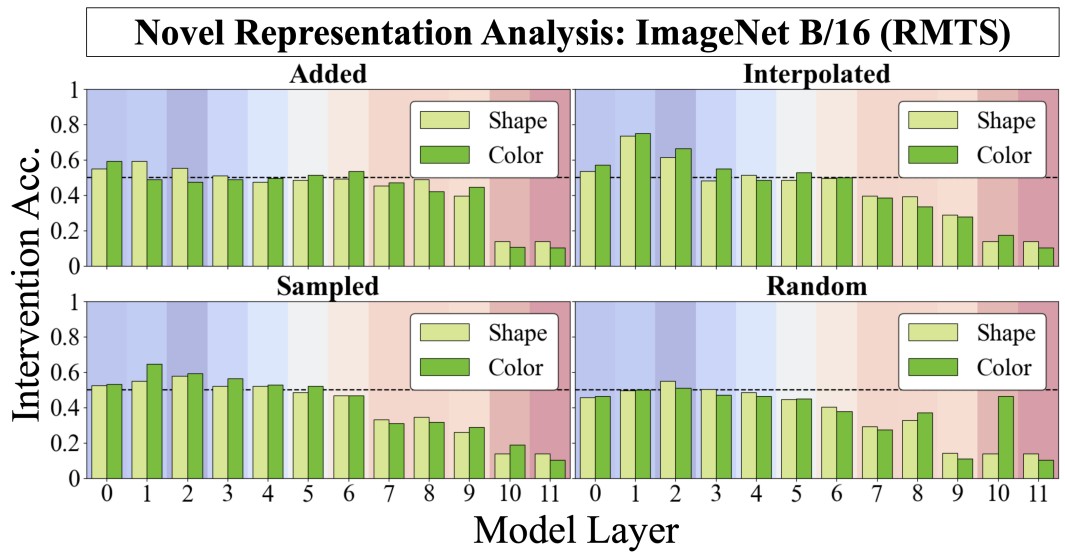

Figure 33: **Novel Representation Analysis for ImageNet ViT-B/16 (RMTS)**.

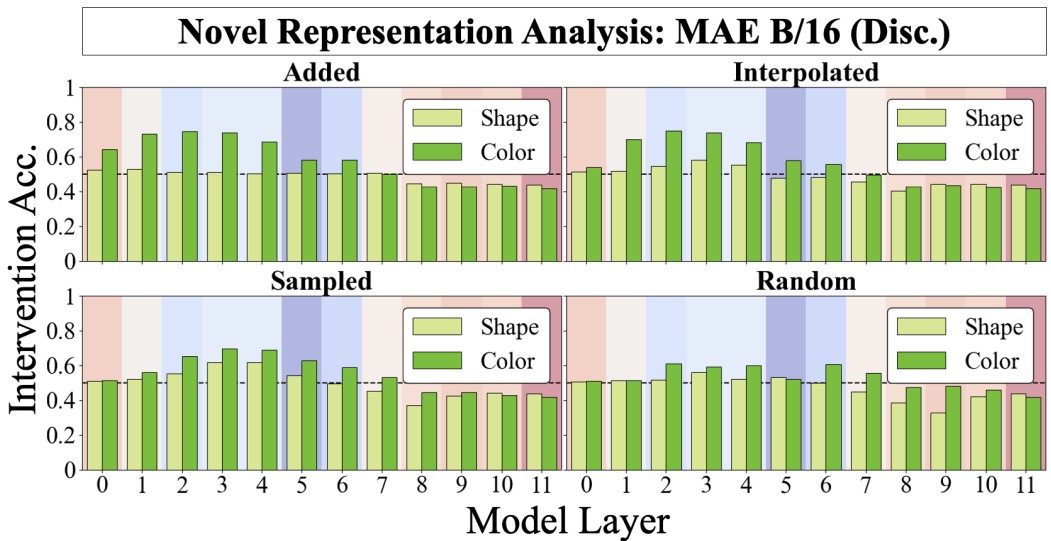

Figure 34: **Novel Representation Analysis for MAE ViT-B/16 (Disc.)**.

To encourage a particular type of attention pattern in a given layer, we first compute the attention head scores (according to Section 3) for a randomly selected subset of either 4, 6, or 8 attention heads in that layer.[10] These scores are then averaged across the layer. The average attention head score is subtracted from 1, which is the maximum possible score for a given attention type (i.e. WO, WP, and BP following Figure 2). This difference averaged across model layers is the pipeline loss term. In the case of within-object attention, an average score of 1 means that each attention head in the selected subset only attends between object tokens within the same object; no other tokens attend to each other. Thus, using the difference between 1 and the current attention head scores as the loss signal encourages the attention heads to assign stronger attention between tokens within objects and weaker attention between all other tokens. The same holds for WP and BP attention. However, the particular forms of the attention patterns are not constrained; for example, in order to maximize the WO attention score in a given layer, models could learn to assign 100% of their attention between two object tokens only (instead of between all four tokens), or from a single object token to itself. This

---

[10]This selection is kept constant throughout training; i.e., the attention heads that receive the pipeline loss signal are randomly chosen before training but do not change throughout training.

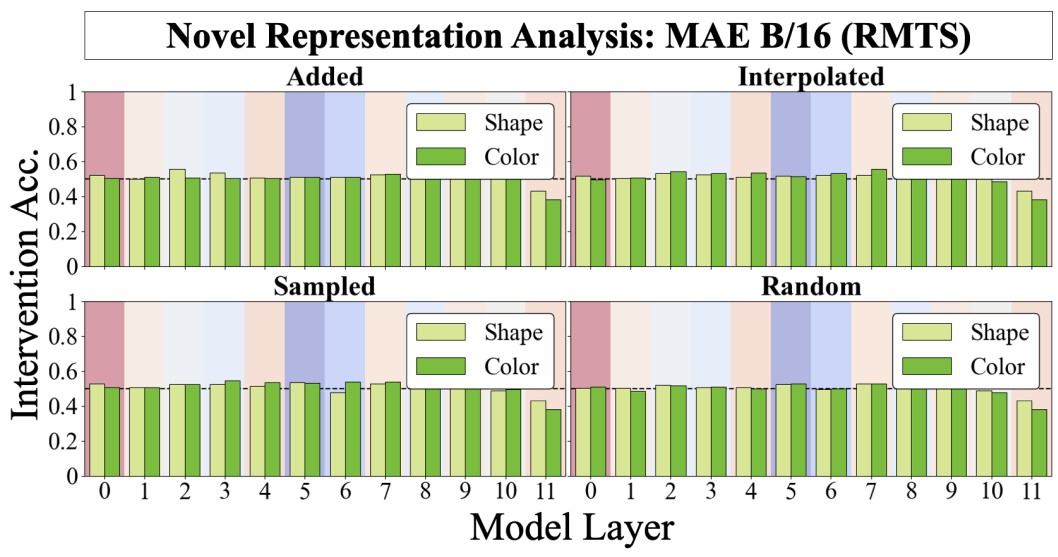

Figure 35: **Novel Representation Analysis for MAE ViT-B/16 (RMTS)**.

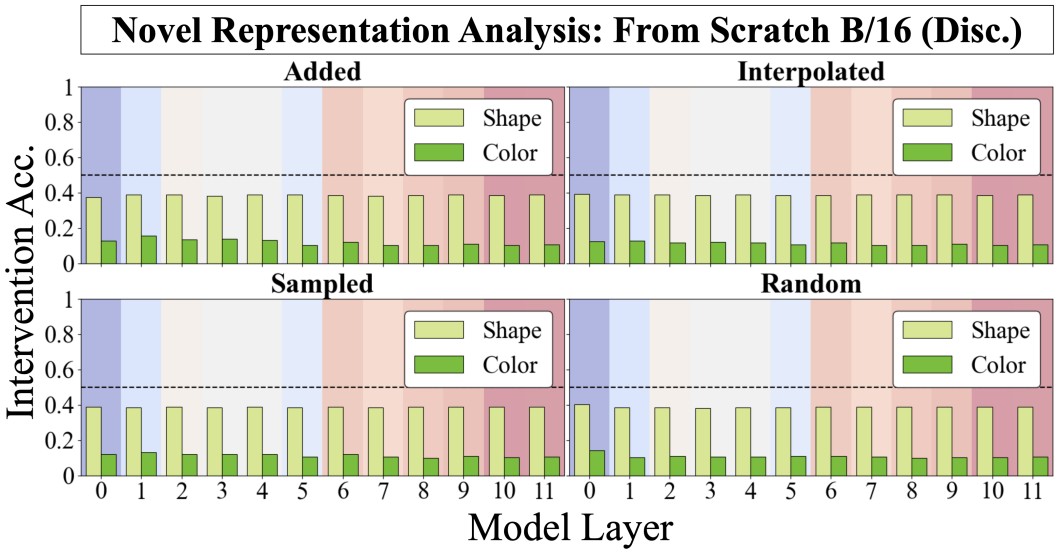

Figure 36: **Novel Representation Analysis for From Scratch ViT-B/16 (Disc.)**.

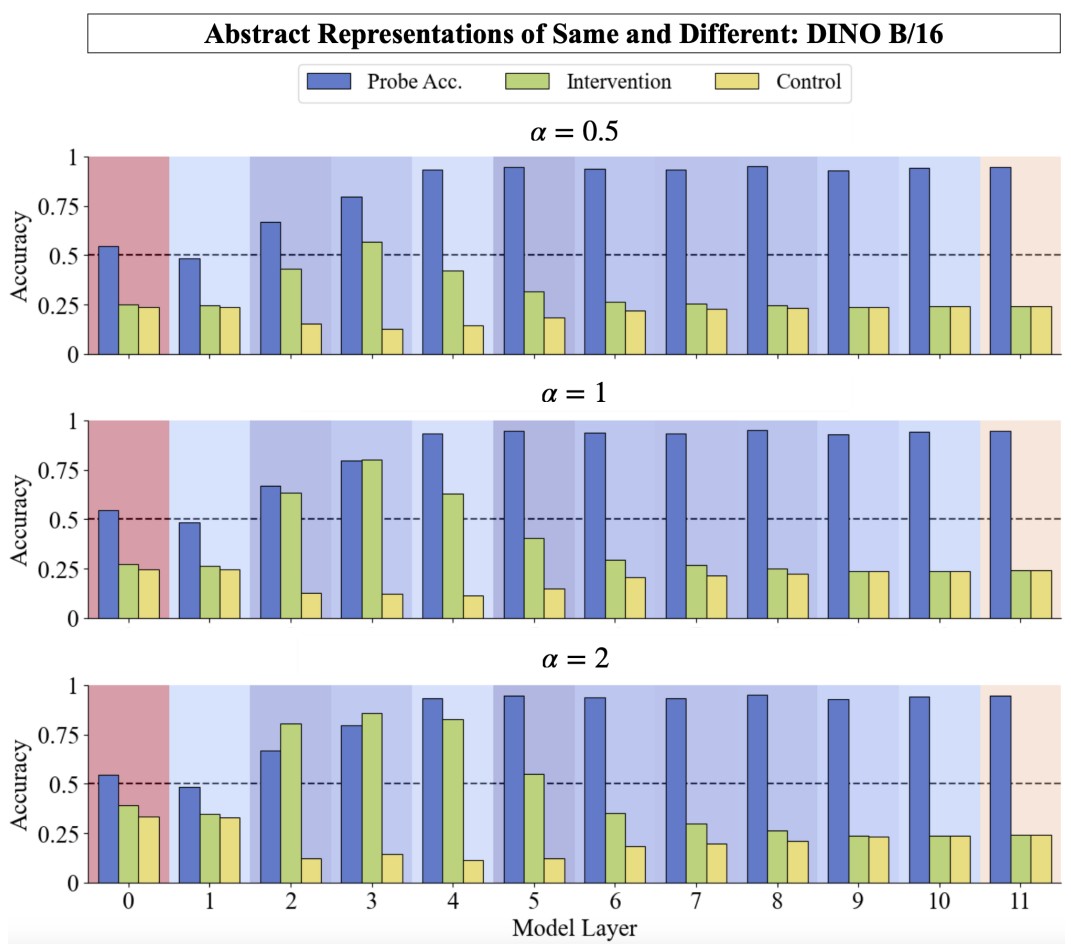

Figure 37: **Scaled linear probe & intervention analysis for DINO ViT-B/16**.

flexibility is inspired by the analysis in Figure 21, which finds that within-object attention patterns can take many different configurations that might serve different purposes in the formation of object representations. The same is true for WP and BP patterns.

## M  Auxiliary Loss Ablations

In this section, we present ablations of the different auxiliary loss functions presented in Section 7. Notably, the pipeline loss consists of two or three modular components, depending on the task. These components correspond to the processing stages that they attempt to induce — within-object processing (WO), within-pair processing (WP), and between-pair processing (BP). For discrimination, we find that either WO or WP losses confer a benefit, but that including both results in the best performance.

For RMTS, we find that including all loss functions once again confers the greatest performance benefit. Notably, we find that ablating either the disentanglement loss or the WP loss completely destroys RMTS performance, whereas ablating WO loss results in a fairly minor drop in performance.

## N  Compute Resources

We employed compute resources at a large academic institution. We scheduled jobs with SLURM. Finetuning models on these relational reasoning tasks using geforce3090 GPUs required approximately 200 GPU-hours of model training. Running DAS over each layer in a model required approximately 250 GPU-hours. The remaining analyses took considerably less time, approximately

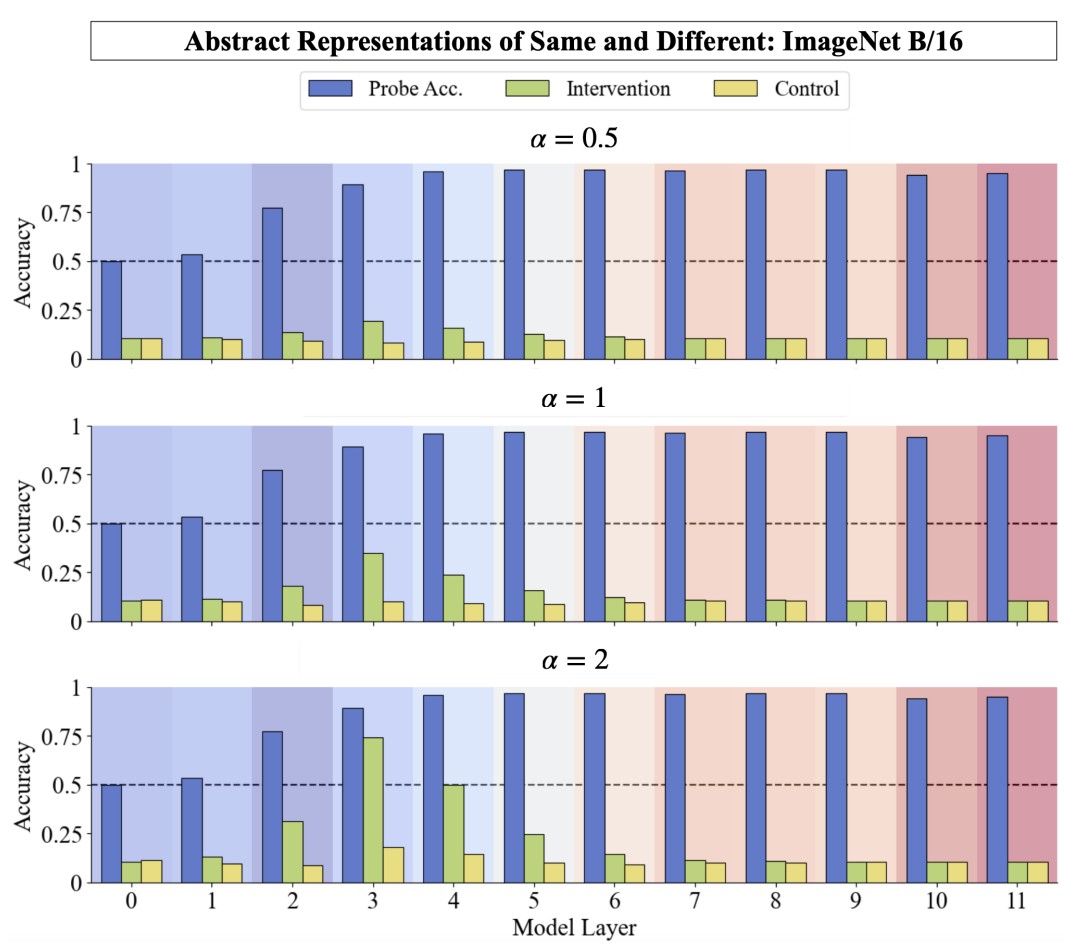

Figure 38: **Scaled linear probe & intervention analysis for ImageNet ViT-B/16**.

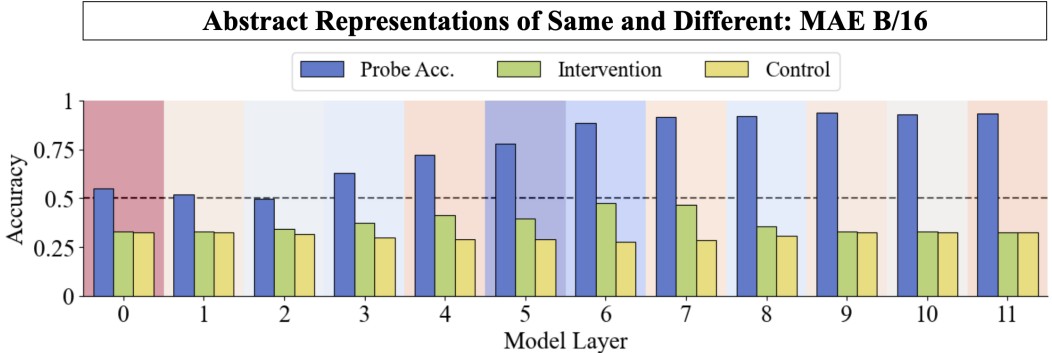

Figure 39: **Linear probe & intervention analysis for MAE ViT-B/16**.

| WO Loss | WP Loss | Train Acc. | Test Acc. | Comp. Acc. |
|---|---|---|---|---|
| – | – | 77.3% | 76.5% | 75.9% |
| ✓ | – | 91.6% | 88.8% | 84.5% |
| – | ✓ | 93.0% | 91.1% | 87.5% |
| ✓ | ✓ | 95.6% (+18.3) | 93.9% (+17.4) | 92.3% (+16.4) |

Table 7: **Performance of ViT-B/16 trained from scratch on the discrimination task with auxiliary losses**.

| Disent. Loss | WO Loss | WP Loss | BP Loss | Test Acc. | Comp. Acc. |
|---|---|---|---|---|---|
| – | – | – | – | 50.1% | 50.0% |
| ✓ | – | – | – | 49.4% | 50.7% |
| – | ✓ | – | – | 49.3% | 51.3% |
| ✓ | ✓ | – | – | 50.0% | 51.3% |
| – | – | ✓ | – | 48.9% | 50.0% |
| ✓ | – | ✓ | – | 85.6% | 68.7% |
| – | – | – | ✓ | 50.0% | 50.1% |
| ✓ | – | – | ✓ | 51.0% | 51.2% |
| – | ✓ | ✓ | – | 61.3% | 50.8% |
| ✓ | ✓ | ✓ | – | 83.2% | 68.8% |
| – | ✓ | – | ✓ | 49.8% | 50.9% |
| ✓ | ✓ | – | ✓ | 49.9% | 50.5% |
| – | – | ✓ | ✓ | 51% | 50.8% |
| ✓ | – | ✓ | ✓ | 87.7% | 76.5% |
| – | ✓ | ✓ | ✓ | 50.1% | 50.1% |
| ✓ | ✓ | ✓ | ✓ | 91.4% | 77.4% |

Table 8: **Performance of ViTs trained from scratch on RMTS with auxiliary losses**.

50 GPU-hours in total. Preliminary analysis and hyperparameter tuning took considerably more time, approximately 2,000 GPU-hours in total. The full research project required approximately 2,500 GPU-hours.

