# OpenReview forum: "Beyond the Doors of Perception: Vision Transformers Represent Relations Between Objects"
_NeurIPS.cc/2024/Conference — NeurIPS 2024 poster_

### Official Review · Reviewer_c3Nf · 2024-07-09

**Soundness:** 3
**Presentation:** 3
**Contribution:** 3
**Rating:** 6
**Confidence:** 5

**Summary:**

The paper aims to answer how Vision Transformers (ViTs) perform tasks requiring visual relational reasoning. The study focuses on two tasks: identity discrimination and relational match-to-sample (RMTS), showing that ViTs process information in two distinct stages: perceptual and relational. Further analyses using mechanistic interpretability approaches were used to validate this results. Overall, this work offered insights into same-different judgement which is intuitive to humans but proven hard for AI.

**Strengths:**

- Similarity judgement task has been long studied in humans and animals, offering insights into perception and cognition, that's why the main question addressed in this study is of high interest for both psychology and AI community
- The analyses included support the main claims
- The boundaries of claims, and the questions remained to be answered were clearly stated in the discussion

**Weaknesses:**

- Related work is poorly presented: Despite the vast literature on both the question and the methodology, only one paragraph was allocated to the related work. There are other works suggesting a two phase processing in transformers (see below for an example) which I think discussing them in the context of this work would help to put the relational reasoning in a broader context.
- Maybe related to point one, but a theoretical explanation supporting the findings, that is, under what conditions these distinct phases appear would make the work stronger.

@misc{cui2024,

      title={A phase transition between positional and semantic learning in a solvable model of dot-product attention},

      author={Hugo Cui and Freya Behrens and Florent Krzakala and Lenka Zdeborová},

      year={2024},

      url={https://arxiv.org/abs/2402.03902},
}

**Questions:**

I am curious about the potential impact of CLIP's text co-training, in contrast to DINO's image-only training, on their respective performances when fine-tuned for same-different tasks. Specifically, I wonder if CLIP's joint training on images and captions forces its embedding space (particularly in the later layers) to deviate from purely visual representations to accommodate textual information as well. Could this difference in training approaches be a significant factor in explaining their divergent behaviors on visual reasoning? In that case, the the perpetual/reasoning phases probably not a property of ViT (as title suggested) but a product of transformer combined with multi-modality.

**Limitations:**

- Related work section is very limited.
- Claims on ViT seem to confined to CLIP, so probably a slight revision of the title and text makes the basis of the claim stronger.

---

> ### Author Rebuttal · Authors · 2024-08-07
>
> Thank you for your thoughtful review! Find point-by-point replies below:
>
> **Weaknesses**:
> 1. “Related work”: We completely agree! We will use the extra space in the final submission to expand our literature review section to include more insights from both the (vast) cognitive science literature and the mechanistic interpretability literature (see global rebuttal #1 for specific topics). We are also happy to include more related work on the processing pipeline in transformers (including the reference you mentioned).
>
> 2. “Theoretical explanation”: We agree, and we are curious to find this out ourselves! While we believe a proper treatment of the learning dynamics that give rise to multi-stage processing in transformers is out of scope for this work, we are highly interested in pursuing this direction in the future. Our current understanding is quoted below and is supported by additional analyses on DINOv2, a vision-only ViT whose pretraining dataset is closer in scale to CLIP than the other models. Despite using a very different training objective, DINOv2 matches CLIP’s performance on discrimination and Relational-Match-to-Sample (RMTS) tasks (99.5% test accuracy on discrimination and 98.2% on RMTS); crucially, it also exhibits similar two-stage processing. We will include these results upon acceptance, though see Figure 1 in the supplemental PDF for an attention pattern analysis on DINOv2.
>
> > “Raghu et al. (2021) finds that models pretrained on more data tend to learn local attention patterns in early layers, followed by global patterns in later layers. This might give CLIP a particular incentive to create local object representations, which are then used in relational operations. Future work might test this hypothesis.”
>
> **Questions**:
> 1. “Multi-modality”: We were very interested in this question as well. As mentioned earlier, we analyze DINOv2 to explicitly test this. While CLIP does show some differences from DINOv2 (see general rebuttal #2 and the response to reviewer NSpf, question #2), DINOv2 largely recapitulates the results from CLIP. This suggests that data scale drives the adoption of two-stage processing rather than multi-modality.
>
> **Citations**:
> 1. Raghu, M., Unterthiner, T., Kornblith, S., Zhang, C., & Dosovitskiy, A. (2021). Do vision transformers see like convolutional neural networks? Advances in neural information processing systems, 34, 12116-12128.

---

> > ### Comment · Reviewer_c3Nf · 2024-08-10
> >
> > Thank you for your response. New experiments on the effect of multi-modality certainly helps support the main claim regarding ViTs (although not quite there yet) and it's encouraging to see authors are willing to address other limitations regarding prior work in the final version. I raided the confidence score. Looking forward to see the improved final version of the work.

---

### Official Review · Reviewer_NSpf · 2024-07-10

**Soundness:** 3
**Presentation:** 2
**Contribution:** 2
**Rating:** 5
**Confidence:** 3

**Summary:**

This paper examines how vision transformers (ViTs) process visual relations, focusing on same-different tasks. The paper finds that pretrained ViTs fine-tuned on these tasks often develop a two-stage processing pipeline: a perceptual stage that extracts object features into separate representations, followed by a relational stage that compares these representations. Using interpretability techniques, they show that the perceptual stage creates disentangled object representations, while the relational stage implements somewhat abstract comparisons. The paper demonstrates that failures in either stage can prevent models from learning generalizable solutions. By analyzing ViTs in terms of these processing stages, the authors suggest we can better understand and improve how models handle relational reasoning tasks.

**Strengths:**

- Addresses an important drawback of supposedly generalist vision models
- Clear demonstration and definition of tasks that the paper focus on (e.g. Figure 1)
- Thorough analysis, both in terms of performance metrics and model mechanism, of results
- Abundant visualization of results
- Constructed dataset and tasks that can benefit future studies of this topic

**Weaknesses:**

- Focus on more toy settings. Lack evaluation on more advanced/real world tasks.
- While the central claim of dividing ViT perception into perception and relation stage is inspired by infant and animal abstract concept (line 71), there's the risk of confirmation bias from this analogy.
- Plots in the paper requires clearer annotations and explanations (e.g. Figure 2 has too many acronyms, and it's difficult to understand what conclusion is for the plot)
- Lacks qualitative example of more intuitively describe how model processes relational examples
- While the abstract claims that understanding the relational reasoning help rectify short comings of existing and futures models (line 19), the paper doesn't discuss much what can be done to improve model relational reasoning (especially empirical experiments for improvements).

**Questions:**

- What would the same analysis methods result when applied on real world images or more complex reasoning tasks such as ARC-AGI?
- Can you compare the mechanistic interpretation result with other text-based interpretability methods such as logic lens?
- How to improve relational reasoning capability in vision models? What are implications of such potential improvements?
- How would success or failure in relational reasoning impact CLIP in terms of embedding's alignment with text descriptions?

**Limitations:**

The author doesn't discuss limitations (although the paper checklist claims that limitations are discussed in Section 8).

---

> ### Author Rebuttal · Authors · 2024-08-07
>
> Thank you for your detailed review! Find point-by-point replies below:
>
> **Weaknesses**:
> 1. “Toy settings”: Fair point! To address this, we created a realistic same-different dataset using 3D models of objects and used it to evaluate our models, similarly to [1]. See Figure 2 in the supplemental PDF for examples of this dataset. We find that CLIP attains a zero-shot test accuracy of 93.9% on this dataset, while all other models attain chance performance. We also find that CLIP (and DINOv2, despite only achieving chance accuracy) exhibits similar two-stage processing on the realistic stimuli without additional fine-tuning on them (albeit with more attention paid to background tokens). This generalizes our results from the toy setting. See Figure 3 in the supplemental PDF for attention pattern analyses on these stimuli. We note that this task is not a perfect analogue to our synthetic setting due to more realistic perceptual variation (i.e. lighting, rotation, occlusion), but hopefully its inclusion can ameliorate concerns about real-world data/tasks!
>
> 2. “Confirmation bias”: This is a reasonable concern. However, we try to let the data speak for themselves. Though our attention pattern analysis might be more qualitative and up for subjective interpretation, our analyses in sections 4 and 5 make clear that there is a marked difference between early and late processing along the lines of what we have described as a “two-stage processing pipeline.”
>
> 3. “Clearer annotations”: We completely agree. Upon acceptance, we plan to use the extra space in the final submission to make this figure larger and include more detailed annotations. We are happy to clarify our interpretation of Figure 2 as well (especially in the caption).
>
> 4. “Qualitative examples”: We have some qualitative examples in Figure 11 (in the appendix), but we have created a new figure that gives much more intuition and have also written a greatly expanded appendix outlining this and Figure 11. See Figure 4 in the supplemental PDF for the new figure.
>
> 5. “Improve model relational reasoning”: In the submission, we attempted to induce relational reasoning by encouraging models to form local, object-level representations using an auxiliary loss in Section 7. However, your point is very well taken, and we had similar thoughts. In light of this, we have devised a new auxiliary loss derived from the attention head scores in Section 3 that explicitly encourages models to exhibit two-stage processing. We hypothesized that this would improve relational reasoning. We find that training randomly-initialized ViTs on the discrimination task with this additional loss term significantly boosts performance (76.5% to 93.9% test accuracy; +17.4). It also boosts the model’s compositional generalization (75.9% to 92.3% accuracy; +16.4). We have promising preliminary results using this loss to improve performance on the RMTS task and will comment on them when ready. This loss is also quite general, and we believe that it can be easily adapted to and possibly improve performance for a wide variety of relations. We will include these new empirical experiments and details about the loss in the main body of the paper upon acceptance.
>
> **Questions**:
> 1. “Real world images”: See response to weakness #1.
>
> 2. “Logit lens”: At your suggestion, we have implemented and run a logit lens analysis on several models, analyzing how different model components contribute to “same” and “different” classifications. The analysis reveals that CLIP seems to process examples in a qualitatively different way compared to the rest of the models, perhaps pointing towards a mechanism for its success on the realistic stimuli. In particular, CLIP appears to make greater use of register tokens in the background compared to other models; it also appears to use the CLS token in non-intuitive ways for intermediate computations (rather than just gradually storing the “same” or “different” decision in CLS across layers, as other models do). We will include this analysis in the appendix, though we leave further investigation to future work.
>
> 3. “Improve relational reasoning”: See response to weakness #5.
>
> 4. “Alignment with text descriptions”: In general, a vision model that cannot perform relational reasoning will not discriminate between closely matched sentences (i.e. “plants surrounding a lightbulb” vs. “a lightbulb surrounding plants”). Indeed, this has been born out in the literature in the form of Winoground [2] among other datasets.
>
> **Citations**:
> 1. Tartaglini, A. R., Feucht, S., Lepori, M. A., Vong, W. K., Lovering, C., Lake, B. M., & Pavlick, E. (2023). Deep neural networks can learn generalizable same-different visual relations. arXiv preprint arXiv:2310.09612.
> 2. Thrush, T., Jiang, R., Bartolo, M., Singh, A., Williams, A., Kiela, D., & Ross, C. (2022). Winoground: Probing vision and language models for visio-linguistic compositionality. Proceedings of the IEEE/CVF Conference on Computer Vision and Pattern Recognition (pp. 5238-5248).

---

> > ### Comment · Reviewer_NSpf · 2024-08-10
> >
> > Thank you for your response. These additional experiments are convincing, and I have raised my score. I have the following additional questions
> > 1. Does stimuli pattern vary across samples and impact model performance?
> > 2. You provide model training with auxiliary loss result in Table 1 of supplementary. What task is the model trained and evaluated on?

---

> > > ### Author Response · Authors · 2024-08-13
> > >
> > > Thank you for the followup questions! We are happy to answer as best we can.
> > >
> > > 1. I am not exactly sure what you mean by this, but the Gaussian noise applied to each object is randomly sampled independently for each image when we create our datasets. So, all blue X's (for example) have different noise patterns in each image. We explored removing the Gaussian noise earlier in this project, and found that this had little effect on downstream performance.
> > >
> > > 2. Table 1 contains results from training models on a discrimination task with the auxiliary loss applied. We use identical training and testing datasets with and without including the auxiliary loss. The training sets contain 32 different shape-color pairs, the test set contains new images with the same 32 pairs, and the compositional dataset contains the remaining held-out pairs. These datasets contain metadata defining the color and shape of each object in the image, and we use this metadata to define our auxiliary loss function. In the prose, we reference an analogous version of this experiment on the RMTS task. Table 4 in Appendix J presents these results.
> > >
> > > We hope this clears things up! Let us know if you have any further followup questions.

---

> > > > ### Comment · Reviewer_NSpf · 2024-08-14
> > > >
> > > > Thank you for your response. It addressed my concerns.

---

### Official Review · Reviewer_DaaT · 2024-07-13

**Soundness:** 2
**Presentation:** 3
**Contribution:** 2
**Rating:** 6
**Confidence:** 3

**Summary:**

The paper uses techniques from mechanistic interpretrability to analyze the algorithms implemented by pretrained ViTs to solve abstract visual reasoning tasks. The authors use two synthetic same-different tasks: discrimination and relational match-to-sample (RMTS), to analyze CLIP pretrained ViTs, DINO pretrained and Imagenet pretrained ViTs, finetuned on these tasks, as well as a ViT trained from scratch. Some pretrained ViTs, especially CLIP pretrained demonstrated a strong perceptual stage in the early layers, disentangling object representations, followed by a relational stage in the later layers, which implements somewhat abstract same-different relations. The authors also demonstrate that the formation of only the perceptual stage is enough to solve simple relational tasks like discrimination, but not enough for RMTS.

**Strengths:**

1. The paper is well written and easy to follow.
2. Interesting use of mechanistic interpretrability to analyze how pretrained ViTs solve same-different tasks: discrimination and relational match-to-sample (RMTS).
3. The paper shows that early layers of CLIP pretrained ViTs demonstrate local attention for within object processing (perceptual stage), whereas the later layers demonstrate global attention for between object processing (relational stage). The two stage processing doesn’t form strongly for the other pretrained models (DINO and ImageNet) especially the relational stage.
4. The perceptual stage is characterized by disentangled object representations (color and shape). Models trained from scratch can be enforced to have disentangled representations using an auxiliary loss which is enough for generalization in simple relational tasks like discrimination but not for RMTS task.

**Weaknesses:**

1. Analysis is limited to simple tasks consisting of simple relations. It remains to be seen if the results would hold up for more tasks involving higher order relations and more complicated relations.

2. The paper doesn’t give a clear and detailed intuition behind why CLIP pretrained ViTs implement the two stage processing more strongly compared to other pretrained ViT models.

**Questions:**

1. According to Fig 2a,c deeper layers don’t seem to be dominated by global heads as WO attention score is higher than WP?

2. Analysis of some models are missing in Fig 2 (e.g DINO Discrimination and Imagenet pretrained ViTs).


3. Can the authors also perform the same analysis for ViTs pretrained using masked autoencoding objective [1]?

4. Are the results in Fig 6 averaged for all models?

[1] - He, K., Chen, X., Xie, S., Li, Y., Dollár, P. and Girshick, R., 2022. Masked autoencoders are scalable vision learners. In Proceedings of the IEEE/CVF conference on computer vision and pattern recognition (pp. 16000-16009).

**Limitations:**

The authors have discussed the limitations in the last section.

---

> ### Author Rebuttal · Authors · 2024-08-07
>
> Thank you for your thoughtful review! Find point-by-point responses below:
>
> **Weaknesses**:
> 1. “Simple relations”: This is fair! With respect to “higher order relations”, we attempted to explore this using the Relational-Match-to-Sample (RMTS) task—an explicitly hierarchical version of the discrimination task. While this goes beyond much of the same-different literature (which focuses on variations of the discrimination task), we do limit our investigation to the same-different relation (rather than other relations) due to its particular significance in the study of abstract concepts in the cognitive sciences, as well as its conceptual simplicity for applying mechanistic interpretability techniques to.
>
> 2. “Detailed intuition”: We agree that this description (pasted below) was somewhat speculative during the submission:
>
> > “Raghu et al. (2021) finds that models pretrained on more data tend to learn local attention patterns in early layers, followed by global patterns in later layers. This might give CLIP a particular incentive to create local object representations, which are then used in relational operations. Future work might test this hypothesis.”
>
> * Since submission, we have rerun our behavioral analysis and attention pattern analysis on a DINOv2 ViT, which is pretrained on about 142M images; this is close in size (in order of magnitude) to CLIP’s pretraining dataset (about 400M images). Despite having a vastly different pretraining objective to CLIP, the DINOv2 model matches CLIP’s performance on both discrimination and RMTS tasks (99.5% test accuracy on discrimination and 98.2% on RMTS). It also exhibits two-stage processing like CLIP. This bolsters our intuition that pretraining data scale (rather than the type of pretraining supervision) is the key. We will include all of these results in the camera ready version. See Figure 1 in the supplemental PDF for supporting figures.
>
> **Questions**:
> 1. “WO attention score is higher than WP”: These attention scores refer to the maximum proportion of attention paid by any given attention head to either within-object, within-pair, between-pair, or background tokens. Though the max for WO is higher than the max for WP in later layers, it is more informative to look at the trend within a given head-type rather than between head types. Notably, we find that the peaks of these scores occur in the expected sequence.
>
> 2. “Analysis of some models are missing”: Thank you for catching this! We will include these graphs in the appendix of the final submission.
>
> 3. “MAE”: Can do! We fine-tune a pretrained ViT-MAE model on the discrimination and RMTS tasks and perform an attention pattern analysis (following Section 3). We find that ViT-MAE achieves very similar performance to ImageNet and DINO ViT. On discrimination, it achieves 98% test accuracy and 94.9% compositional generalization accuracy. On RMTS, it achieves 93.4% test accuracy (interestingly, somewhat higher than ImageNet and DINO) and 85.3% compositional generalization accuracy. Its attention patterns do not demonstrate two-stage processing like CLIP or DINOv2; instead, the local and global heads are mixed throughout the layers. We will add an attention pattern analysis on MAE to the appendix of the paper.
>
> 4. “Figure 6”: No—each data point corresponds to a different model, which achieves a different maximum disentanglement score. We will revise the caption for clarity.
>
> **Citations**:
> 1. Raghu, M., Unterthiner, T., Kornblith, S., Zhang, C., & Dosovitskiy, A. (2021). Do vision transformers see like convolutional neural networks? Advances in neural information processing systems, 34, 12116-12128.

---

> > ### Comment · Reviewer_DaaT · 2024-08-14
> > **Official comment by Reviewer DaaT**
> >
> > Thank you for the detailed rebuttal, which has addressed some of my concerns. I have increased the rating to 6.

---

### Official Review · Reviewer_UKTX · 2024-07-13

**Soundness:** 2
**Presentation:** 2
**Contribution:** 2
**Rating:** 5
**Confidence:** 3

**Summary:**

This work studies ViTs' learning behavior with relational tasks by experimenting on 2 same-different tasks: discrimination and RMTS tasks.  And the authors propose a dataset to analyze. And discovers that there are 2 stages of attention processing of CLIP ViTs by attention scores from patches to other patches. They characterize 2 stages, perceptual stage and relational stage.

**Strengths:**

This paper defined a novel avenue with inspiration from mechanistic interpretability to understand the working mechanism of CLIP ViTs. And proposed approaches to study them with attention analyses.

**Weaknesses:**

1. I believe readers in this track would generally not be experts in concepts of the proposed area of studies, the paper should include more literature review for context upfront.

2. I find the writing and the flow of the paper hard to follow, maybe adding flow-chart would help improve readability.

3. The dataset is not well-discussed.

4. I think there is a slight mismatch between claimed study and experiments. The title suggest the study centers around ViTs but use CLIP ViTs, leaving me confused if the points made are the same on the original ViTs.

**Questions:**

1. ln 23-24, the motivation "little breakthrough progress on complex tasks involving relations between visual entities..." is that true? What about the recent breakthroughs or datasets of vision-language models, Llava, GPT4, etc. Please give more explanations.

2. ln 41, please define what are "algorithms" learned by ViT, and are there algorithms learned by other architectures, CNNs, LSTMs, etc.?

3. Are there no existing datasets that can study the relations of visual entities? I mean in general, not the RMTS, etc.

3. The motivation of using CLIP ViTs for study is not clear. Since part of the motivation is from infant learning, why not use more human-like CNN like ConvNext? Also, CLIP ViT is different from ViT as well.

4. Vaguely remember [1] discussed about using better disentangled model for generalization and color, are there new insights from Sec 6?

5. With the 2 stage processing, what would you suggest the architecture to change to build a more "robust" version?
---------
[1] Better may not be fairer? [Chiu, et. al., 2023]

**Limitations:**

Yes.

---

> ### Author Rebuttal · Authors · 2024-08-07
>
> Thank you for your thoughtful comments and questions. Find point by point responses below:
>
> **Weaknesses**:
> 1. “...Literature review for context…”: We completely agree. Upon acceptance, we will use the extra space to expand our literature review section to include more insights from both the (vast) cognitive science literature and the mechanistic interpretability literature (see global rebuttal #1 for specific topics).
>
> 2. “Flow of the paper”: We are happy to add more signposting throughout the sections. Roughly, the paper is structured such that Section 3 motivates the investigation of two stages of processing, Sections 4 and 5 respectively investigate these stages in greater detail, and Sections 6 and 7 discuss the implications of some of our findings on downstream performance. We are happy to state the structure of the paper in the introduction!
>
> 3. “Dataset is not well discussed”: We have written a vastly extended appendix detailing the dataset and its construction, which we will include in the final submission. We are also happy to answer any questions about the dataset during the discussion.
>
> 4. “CLIP ViTs”: In this work, we investigate abstract relations in a variety of ViTs. In particular, we study ViTs that are pretrained with different kinds of supervision, e.g. CLIP, DINO, and ImageNet. However, as CLIP is the only type of pretraining that achieves near perfect test accuracy for our tasks, we focus on it for the majority of the paper. Since submission, we have added DINOv2 to our analyses, finding that it solves both discrimination and Relational-Match-to-Sample (RMTS) tasks with near perfect test accuracy (99.5% test accuracy on discrimination and 98.2% on RMTS); importantly, it also exhibits two distinct stages of processing similar to those we observe in CLIP. This generalizes our findings beyond CLIP and bolsters our intuition that pretraining data scale (rather than something particular to CLIP) provides the visual representations needed to solve these tasks. We will include these results in a new appendix to the paper and reference them throughout the main body.
>
> **Questions**:
> 1. “Little progress”: Great point—we should be more specific. We mean that across a variety of tightly controlled benchmarks, vision models tend to struggle with tasks that involve relations [3, 4], especially compared to tasks that involve semantic computations. Additionally, the models that you mention have demonstrated some progress in processing visual relations, but there are still shortcomings to be addressed. We are happy to add these references to the final manuscript in an updated related work section.
>
> 2. “Algorithms”: This terminology is borrowed partially from the mechanistic interpretability literature (i.e. in conceptualizing neural networks as implementing interpretable algorithms), and partially from the cognitive science literature (i.e. Marr’s levels of analysis). When we use the word “algorithm”, we are referring to the series of representations generated by a model to solve a particular task.
>
> 3. “Existing datasets”: There are plenty of datasets used to study visual relations! For example: SVRT [1] and CVRT [5]. However, these datasets are not controlled enough to enable us to adapt state-of-the-art techniques from language model mechanistic interpretability for ViTs. Our dataset is explicitly constructed to enable these techniques by e.g. consistently aligning objects within the bounds of ViT patches. Our new dataset appendix details our design choices.
>
> 4. “Motivation of using CLIP ViT”: As mentioned above (weakness #4), we use many different ViTs. Your point about CNNs being more human-like is well taken! We use ViTs because prior work has demonstrated that they can solve same-different tasks in a robust fashion [2]; they also enable us to use techniques from mechanistic interpretability that have previously been developed for Transformer language models (DAS, attention analysis, linear probing-based causal interventions, logit lens). Since CNNs lack tokens, it is not clear whether it is possible to apply these techniques to them.
>
> 5. “Disentangled model for generalization”: We are happy to include this reference in the main body of the paper. Notably, our work demonstrates the benefits of disentangled representations for compositional generalization (rather than standard generalization) in a vastly different setting!
>
> 6. “Build a more robust version”: We were very motivated by this question as well! Since submission, we have implemented a new auxiliary loss function derived from the attention head scores in Section 3 that encourages the model to adopt two-stage processing. We find that this helps a randomly-initialized ViT achieve significantly better downstream performance on the discrimination task (76.5% to 93.9% test accuracy; +17.4). It also significantly boosts compositional generalization accuracy for discrimination (75.9% to 92.3% accuracy; +16.4). We currently have promising preliminary results using this loss to improve performance on the RMTS task and will comment on them when they are ready. This loss is also very general and could possibly be used in the future to improve relational reasoning for a wide variety of visual relations. We will include these results and more details about the loss in the main body of the paper upon submission.
>
> **Citations**:
> 1. Fleuret, F., et al. (2011). Comparing machines and humans on a visual categorization test.
> 2. Tartaglini, A. R., et al.  (2023). Deep neural networks can learn generalizable same-different visual relations.
> 3. Thrush, T. et al. (2022). Winoground: Probing vision and language models for visio-linguistic compositionality.
> 4. Zeng, Y., et al. (2024). Investigating Compositional Challenges in Vision-Language Models for Visual Grounding.
> 5. Zerroug, A., et al. (2022). A benchmark for compositional visual reasoning.

---

> > ### Comment · Reviewer_UKTX · 2024-08-13
> >
> > I would like to thank the authors for their rebuttal and acknowledge that some concerns have been addressed.

---

### Author Rebuttal · Authors · 2024-08-07

We thank all of the reviewers for leaving thoughtful, high-quality comments and questions on our manuscript. Here, we address some common themes found in multiple reviews, and also list all of the additional analyses that we have performed (or are planning to perform) to address any outstanding concerns.

1. **Literature Review**: Several reviewers noted that our literature review was somewhat sparse. We agree, and we plan to greatly expand it using the additional page conferred upon acceptance. In particular, we will add an expanded discussion of prior work evaluating visual relations in computer vision models, a discussion of interpretability work identifying processing pipelines in transformers, and an expanded discussion of abstract visual reasoning in humans.

2. **Synthetic Task Concerns**: Several reviewers raise concerns about the relative simplicity of our datasets. We first note that our dataset is constructed to enable the use of state-of-the-art mechanistic interpretability techniques from NLP on a ViT (some for the first time, to our knowledge!). As in the language domain, these techniques require a high degree of dataset control, and they tend to trade task and data complexity for the ability to make precise interpretations. However, we understand these concerns and have run a behavioral evaluation on a realistic same-different dataset (made using 3D models of objects) to demonstrate the generalizability of our findings. We discover that CLIP attains a zero-shot test accuracy of 93.9% on the realistic dataset; all other models achieve only chance accuracy. Furthermore, CLIP exhibits the same two-stage processing on these stimuli without any additional training on them (albeit with greater attention paid to background tokens throughout the model). See Figure 2 in the supplemental PDF for example stimuli from the realistic dataset and Figure 3 for attention head scores for CLIP (and a new model, DINOv2; see reply #3 below) on these stimuli.

3. **CLIP ViTs vs. Other ViTs**: Several reviewers note that our analyses mainly focus on CLIP ViTs rather than ViTs in general. In the submission, this is simply because CLIP attains the best performance on our tasks. Thus, since CLIP yields the largest number of accurate responses, focusing on it simplified the application of the mechanistic interpretability techniques we use. However, this raises the question: is two-stage processing a result of pretraining data scale (as we suggest) or multimodal pretraining? To address this, we have since run additional analyses on DINOv2 [1], another ViT that is pretrained without linguistic supervision on a dataset of similar size (in order of magnitude) to CLIP (~142M images vs. CLIP’s ~400M). We find that DINOv2 performs as well as CLIP on both discrimination and RMTS tasks (99.5% test accuracy on discrimination and 98.2% on RMTS); it also demonstrates two-stage processing like CLIP. We will include these results in the paper upon acceptance; we have also included the attention pattern analysis for DINOv2 in Figure 1 in the supplemental PDF.

4. **How to Improve Models**: Several reviewers ask how our results might lead to models that achieve stronger relational reasoning skills. We had the same question! Since submission, we have derived a new auxiliary loss from our attention head scores in Section 3 that encourages models to exhibit the two-stage processing found in CLIP (and DINOv2). We find that introducing this loss during training significantly boosts the performance of a randomly-initialized ViT on the discrimination task (76.5% to 93.9% test accuracy; +17.4)—interestingly, it boosts the model’s compositional generalization as well (75.9% to 92.3% accuracy; +16.4). See Table 1 in the supplemental PDF for more results. We have also obtained promising preliminary results using this loss to improve performance on the Relational-Match-to-Sample (RMTS) task and will comment with those results when ready. The formulation of the loss is general (i.e. not specific to same vs. different), so we believe that it could potentially be useful for future work looking to improve performance on other visual relations. We plan to include these empirical results and more details about the loss in the final submission.

**List of additional analyses**:
- DINOv2 behavioral results and attention analysis (will include Section 4, 5, & 6 analyses on DINOv2 with final submission); Figure 1 in supplemental PDF
- Realistic same-different evaluations; Figures 2 and 3 in supplemental PDF
- Experiments using an attention pattern loss to improve relational reasoning; Table 1 in supplemental PDF
- Logit lens implementation and analyses
- ViT-MAE behavioral and attention analysis

**Citations**:
1. Oquab, M., Darcet, T., Moutakanni, T., Vo, H., Szafraniec, M., Khalidov, V., ... & Bojanowski, P. (2023). Dinov2: Learning robust visual features without supervision. arXiv preprint arXiv:2304.07193.

---

> ### Author Response · Authors · 2024-08-13
> **Update on Point 4: How to Improve Models**
>
> We have continued to experiment with integrating attention-based losses to improve model performance in the RMTS task. We have found that combining an attention based auxiliary loss + the shape/color loss described in the manuscript improves performance dramatically for a randomly-initialized model trained on the RMTS task. Specifically, ViT B/16 accuracy improves from ~50% train, test, and compositional generalization accuracy **without auxiliary losses** to 95% train accuracy (+45%) , 91% validation accuracy  (+41%) , and 77.4% compositional generalization accuracy (+27%) **with both attention and shape/color losses**.
>
> Notably, merely including the shape/color auxiliary loss results in far weaker performance improvements, at 54% train, test, and compositional generalization accuracy. These results are already present in Table 4 of the manuscript.
>
> In summary, this experiment further supports our main conclusion: that both multi-stage relational processing and well-structured object representations are required to solve the RMTS task. Inducing both of these factors is possible using straightforward auxiliary losses during training.

---

### Decision · Program_Chairs · 2024-09-25

**Decision:**

Accept (poster)

**Comment:**

The paper investigates emerging capabilities of visual reasoning developed within Visual Transformers. The reviewers found the paper interesting, with novel approach towards mechanistic interpretability. They also appreciated the clarity of presentation. Several weaknesses should be addressed though before the paper is published, namely the extended literature review, confusion about using CLIP ViTs or others, as well as the intuitive explanations how model processes relational examples.